# Atmospheric visibility inferred from continuous-wave Doppler wind lidar

Manuel Queißer[1], Michael Harris[1], Steven Knoop[2]

[1] ZX Lidars, The Old Barns, Fairoaks Farm, Hollybush, Ledbury, HR8 1EU, U.K.

[2] Royal Netherlands Meteorological Institute (KNMI), Utrechtseweg 297, 3731 GA, De Bilt, the Netherlands

*Correspondence to*: Manuel Queißer (manuelqueisser@web.de)

**Abstract.** Atmospheric visibility, or meteorological optical range (MOR), is governed by light extinction by aerosols. State-of-the-art visibility sensors, such as employed in meteorological observatories and airports, infer MOR by either measuring transmittance or scattering. While these sensors yield robust measurements with reasonable accuracy (10% to 20%), they measure in situ. MOR from these sensors may thus not be representative of MOR further away, for example, under conditions with stratified aerosol types. This includes off-shore sites near the sea surface during conditions with advection fog, sea spray or mist. Elastic backscatter lidar can be used to measure light extinction and has previously demonstrated to be a powerful method to infer visibility. Lidar can measure visibility not just near the instrument, but further away (remotely) and single-ended, whilst capable of measuring profiles of MOR along atmospheric slant paths. Continuous-wave (CW) Doppler wind lidar systems make up one of the most widespread type of elastic backscatter lidar and are typically used in wind resource assessment. Using these existing platforms for remote and single-ended measurement of MOR-profiles could allow for new and valuable applications. However, the low light extinction associated with this type of lidar excludes the use of the extinction coefficient for MOR retrieval, but leaves the backscatter coefficient as a possible proxy for MOR, though with an accuracy expected to be inferior to the former method. We analysed backscatter data from CW wind lidar and co-measured MOR from visibility sensors from two campaigns (Cabauw, Netherlands and Pershore, United Kingdom) and found backscatter from CW wind lidar to be a viable proxy of MOR if calibrated against a visibility sensor. The expected accuracy of the method is low and of order of few kilometres. This means MOR from CW wind lidar could be used in safety uncritical problems, such as assessment of visibility of man-made objects, including wind turbines.

## 1 Introduction

Visibility is how well we can see something. More specifically, atmospheric visibility is the maximum horizontal distance an object can be seen through the atmosphere with the naked eye. Visibility is traditionally estimated doing exactly that, namely by measuring the maximum distance a dark object with a suitable size can be seen on the horizon against the surrounding sky. The visibility of a distant object is a function of several factors, including the object's colour, the angle of the sun and Earth's

curvature. However, at constant illumination by the sun, the governing physical mechanism of visibility is the extinction of light by scattering through aerosols suspended in the atmosphere. This leads to the definition of meteorological optical range (MOR), which quantifies that part of visibility that is caused by light extinction. The MOR therefore provides a quantifiable estimate of atmospheric visibility. The world meteorological organization defines MOR as the length of path in the atmosphere required to reduce the luminous flux in a collimated beam from an incandescent lamp, at a color temperature of 2700 K, to 5

per cent of its original value, the luminous flux being evaluated by means of the photometric luminosity function of the International Commission on Illumination (Jones et al., 1990). Note that a blackbody temperature of 2700 K corresponds to about 1100 nm. For daylight conditions, MOR is commonly defined as the horizontal range for which the light intensity contrast between the object and the surrounding sky is either $C_t$=2%, or, as later suggested, 5% (Middleton 1947, Dabberdt and Eigsti 1981; Nebuloni, 2005). Adopting the well-known Bouguer–Lambert–Beer law and $C_t$=5%, MOR can be written

as

$$MOR = \frac{-\ln(C_t)}{\sigma} \approx \frac{3}{\sigma}, \tag{1}$$

where $\sigma$ is the atmospheric extinction coefficient (in units of m$^{-1}$). Hence, by measuring the light extinction $\sigma$, MOR can be derived. In practice, MOR is evaluated at a wavelength of 550 nm, close to the human eye's sensitivity maximum (Nebuloni, 2005). In the rest of the paper the terms visibility and MOR are used interchangeably.

The traditional method of estimating MOR using human visual observation of targets of well-defined distance and albedo delivers values reasonably close to MOR from Eq. (1) and has the advantage that the visibility does not need to be translated into a quantity perceptible to humans. The WMO recommends to measure visibility at a height ca. 2 m above ground, close to human line of sight (Jones et al., 1990).

This is also the height at which state of the art visibility sensors, or visiometers, determine MOR, using Eq. 1. These

visiometers either measure atmospheric transmittance along an optical path of fixed distance or the intensity of forward scattered light, from which the extinction coefficient can be retrieved (Crosby, 2003; Werner et al. 2005). One main difference between these two approaches is that unlike the transmissivity metre, a sensor measuring forward scattering needs a very small optical path only (~10 cm), i.e. it measures in situ, which makes alignment and maintenance easy. Arguably, this risks to sample a portion of the sky that is not representative of the wider atmospheric conditions. Both of these approaches are bound

to measure visibility at the height they are mounted at, usually few metres above ground level (agl) and both are double-ended, meaning the receiver is located at the opposite end of the optical transmitter.

The aerosol number density, and thus the visibility, is generally a function of height agl. Especially offshore, a strong vertical stratification at the sea/atmosphere interface due to condensation caused by advection is to be expected. A visibility sensor mounted a few metres above the sea surface (e.g. at the beach or on a buoy) could be immersed in a layer of spray or

haze, giving a biased visibility reading. An effective visibility measured vertically, rather than horizontally, considering aerosol stratification effects in the boundary layer, or, more generally, a slant optical range (SOR, Werner et al., 2005) may be more desirable for some applications (Fig. 1). For large features, such as mountains or man-made structures exceeding 100 m height, it may be more beneficial to measure an effective visibility at or near the actual line of sight between the observer's eye and

the feature, which would be significantly higher above ground than only a few metres. For example, the height at half distance

for a 100 m high object located 10 km away from the observer would be ~50 m agl.

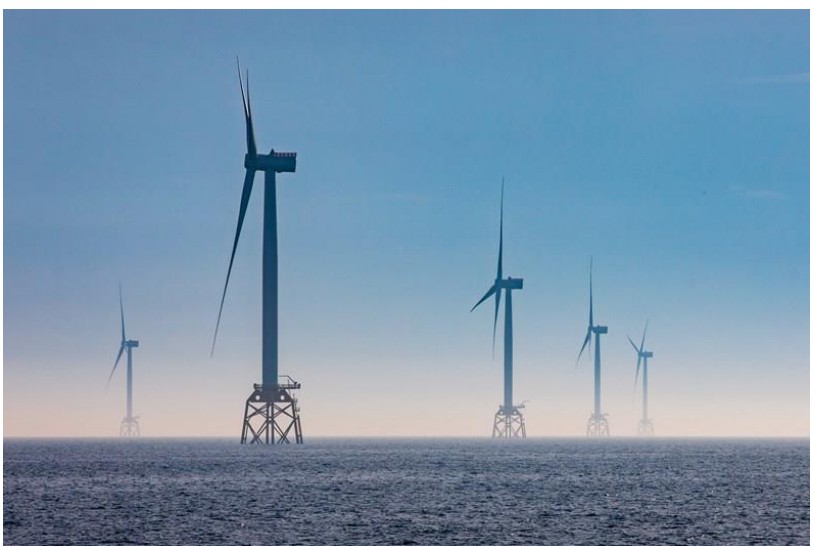

**Figure 1. Visibility of wind turbines offshore. The wind turbines are forming an optical contrast against a clear sky with moderate mist near the sea surface, which reduces visibility for the lower parts of the wind turbines. Photo from: SSE Renewables.**

Light detection and ranging (lidar) allows a spatially integrated measurement of atmospheric visibility, in a single

ended manner and remotely, with ranges of hundreds to kilometres away from the lidar. MOR measured with pulsed elastic backscatter lidar, including ceilometers, has shown good agreement with standard in-situ sensors, such as transmissometer and visibility sensors (Werner et al., 2005; Pantazis et al., 2017; Hongda et al., 2017; Hu et al., 2021). Shang et al. (2017) compared visibility from a pulsed backscatter lidar with visibility from a visibility sensor and found excellent agreement. Lidar is ideal for atmospheric sounding, which means it is able to measure visibility not only at a single location remotely, but at several

points along the lidar line of sight, yielding a visibility profile. The return power of pulsed backscatter lidar is a function of both the backscatter coefficient and the atmospheric extinction coefficient. Therefore, the lidar return signal contains information about the atmospheric extinction and hence the visibility (Eq. 1). The retrieval of the visibility is based on solving the lidar equation for both the atmospheric extinction coefficient and the backscatter coefficient. Retrieval techniques include inversion (e.g. Werner et al., 2005) and iterative measurements starting from an aerosol free reference height (Pantazis et al.,

80    2017).

The question arises, whether MOR can be retrieved from only one of the two parameters, backscatter coefficient or extinction coefficient. Single-ended retrieval of visibility inferred from the backscatter coefficient only has successfully been demonstrated (Curcio and Knestrick, 1958; Vogt 1968), but, as opposed to lidar, by using polychromatic light. The backscatter coefficient has a higher sensitivity to the size of the aerosols along the beam path and hence to the aerosol size distribution

(SD) than the extinction coefficient (Twomey and Howell, 1965). These fluctuations are smoothed considerably when the emitted photons are polychromatic, leading to accuracies of the order of 20% (Twomey and Howell, 1965), which is

comparable to forward scattering visiometers (Biral SWS Series user manual, 2017; Cambell Scientific CS125 user manual, 2016; Crosby, 2003). Vogt (1968) found that the backscatter coefficient is useful to infer visibility, but it needs to be calibrated against a visibility sensor in an atmosphere similar (similar mean aerosol SD) to the one of its intended use. Monochromatic light, as commonly used in lidar, was found to yield a poorer, more scattered correlation between backscattered light intensity and visibility, with an associated scattering around 50% of backscattered intensity for a given visibility (Twomey and Howell, 1965).

Amplitude modulated direct detection continuous wave (CW) lidar has also been proposed and demonstrated to measure MOR by relating it to the phase shift between transmitted and received photons (Schappert 1971; Kreid 1976; Button, 1977). Provided a homogenous atmosphere, the phase shift depends only on the extinction coefficient, avoiding the dependence on the backscatter coefficient (Kreid 1976).

Coherent CW Doppler wind lidar has not been used yet to measure visibility. Thousands of these lidars are being deployed worldwide, especially in wind resource assessment campaigns (Emeis et al., 2007). Coherent CW wind lidars are designed to measure wind field quantities such as speed, direction and turbulence index at heights between tens and few hundreds of metres. Similar to other backscatter lidar systems, a CW Doppler wind lidar measures the light power returned from atmospheric aerosols moving with the wind stream lines. Therefore, CW wind lidar is in principle able to measure visibility, in particular SOR, single-ended and remotely and at different heights (profiling). However, the influence of atmospheric extinction scales with measurement range and wavelength (light with longer wavelength is scattered less).

As opposed to pulsed aerosol lidar described above, CW wind lidar has a lower measurement range. In addition, CW wind lidar operates in the short wave infrared region close to 1550 nm, which is a factor of 1.5 to 3 longer than for typical aerosol lidar and ceilometer systems (Werner et al., 2005; Gasteiger et al., 2011; Navas-Guzmán et al., 2013; Shibata et al., 2018). Compared with pulsed aerosol lidar, at normal working ranges (up to 300 m), the return signal of a CW wind lidar is thus not sensitive to atmospheric extinction, but is practically governed by the backscatter coefficient only. This is illustrated in the following example. Assuming a common visibility of 10 km (slightly hazy), the maximum extinction (i.e. at range 300 m) would only be $e^{-600\,m\,5\times10^{-5}m^{-1}} = 0.97$. This leaves only the backscatter coefficient (henceforth termed backscatter) as the most obvious proxy of visibility of a CW wind lidar.

The aforementioned studies do suggest a correlation between visibility and backscatter, but at relatively poor accuracy, since CW wind lidar is inherently monochromatic. The accuracy may, however, suffice for non-safety critical applications, such as estimates of minimum or maximum visibility. An example would be the visibility of manmade structures such as wind turbines, towers or bridges. A relation between signal strength of CW wind lidar and visibility is evident from everyday operation, but has not been quantified yet. Thousands of wind lidar systems already in operation worldwide could be used to yield a useful remote measure of visibility add no extra cost. This could open new applications of these existing systems in commercial or scientific applications. Systems used for scientific or observational use could provide for a denser, more global data set of visibility. In the present case study, we use datasets from two different measurement campaigns, in Cabauw (Netherlands) and Pershore (UK), to assess if backscatter from CW wind lidar can be used to retrieve meaningful

estimates of visibility. The wind lidar used in both campaigns and the visibility sensors are briefly detailed. After that, two methods to retrieve visibility from backscatter are described. Results from the two methods are presented thereafter and discussed before conclusions are drawn.

## 2 Method and Material

### 2.1 CW wind lidar operating principles

Both vertical profiling wind lidars used in the two measurement campaigns are of type ZX 300 (ZX Lidars, UK, formerly ZephIR Lidar). An overview of some of the properties and settings of the ZX 300 as deployed at Cabauw is given in Table 1. The ZX 300 is a homodyne coherent detection CW focusing wind lidar. The laser beam is transmitted through a constantly rotating prism (wedge) to perform a so-called velocity azimuth display (VAD) scan with a scanning cone angle of ~30° (with respect to zenith). Up to 10 measurement heights can be configured, in addition to a pre-fixed height of 39 m agl, which are permuted through in descending order. Once a measurement height is set by focusing the laser beam, the focus performs a circular scan for 1-second, split into 50 points with ~20 ms integration time, separated by (360/50)°. The 50 measurements are used to reconstruct the vertical and horizontal wind speed components. The optimum height range is 10–200 m above the instrument, although higher heights can be set in the software. By virtue of the geometrical focusing, the probe length increases quadratically with measurement distance (height): at 10 m height above the instrument, the -3dB probe length is 0.07 m, whereas at 200 m it is 30 m. CW focusing wind lidars can be sensitive to clouds that are above the maximum range, as the contribution to the Doppler signal from clouds in the tail of the laser pulse profile can be comparable to the aerosol signal at the preselected focusing height (Smith et al., 2006). A cloud removal algorithm is used to correct for this effect by removing Doppler signal biased by cloud returns, which involves a measurement at an additional higher altitude. For a CW coherent wind lidar, such as the ZX 300, the time-averaged optical signal power $P_S$ backscattered by aerosols into the receiving telescope is given to a good approximation by (Harris et al., 2001)

$$P_S = \pi P_T \beta \lambda, \tag{2}$$

where $P_T$ is the transmitted laser power and $\beta$ is the atmospheric backscatter coefficient, and $\lambda$ is the laser wavelength. Note the difference to pulsed lidars and ceilometers. Equation (2) contains no dependence on either the focus range or the system aperture size. Moreover, the system is monostatic. That means, the typical measurement range of a CW lidar is 0 to few hundred meters with full overlap of transmitted and received beam, which is different to pulsed, bistatic lidars, which have limited overlap in the near field. With a typical value of $10^{-8}$ m$^{-1}$ sr$^{-1}$ for $\beta$ in relatively clear boundary layer air a transmitted power $P_T$ of typically ~1 W and $\lambda$ ~1.5 $\mu$m, the received power $P_S$ derived from Equation (2) is of order $5\times10^{-14}$ W (50 femtowatts) only. Assuming shot noise limited detection, the carrier-to-noise-ratio of a coherent CW lidar is given by (Harris et al., 2001)

$$CNR = \frac{\eta P_S}{\frac{hc}{\lambda} \Delta v [1 + D(v)]}, \qquad (3)$$

where $\eta$ is the quantum efficiency of the detector, $h$ is Planck's constant, $c$ the speed of light, $\Delta v$ the coherent detection bandwidth and $D(v)$ is the detector dark noise spectral density. Combining Eqns. (2) and (3) yields the relationship used to retrieve the backscatter coefficient:

$$\beta = \frac{\left(\frac{hc}{\lambda}\right)\Delta v \, [1 + D(v)] NB}{\pi \eta \lambda P_t} \equiv Z \, B, \qquad (4)$$

with $CNR \equiv NB$, where $N$ is the number of bins of the discrete Doppler spectrum of the lidar, $B$ is the average heterodyne signal power spectral density per bin and computed as

$$B = \frac{1}{N}\sum_{i=0}^{N-1} P_i, \qquad (5)$$

where $P_i$ is the power spectral density of bin $i$. Plugging in numbers typical for a ZX 300 yields $Z = 1.3 \times 10^{-6}$ m$^{-1}$ sr$^{-1}$, which is a constant for a given transmitted optical power. If not written otherwise, backscatter values are henceforth written in units of $10^{-6}$ m$^{-1}$ sr$^{-1}$.

**Table 1. Key parameters of a ZX 300 wind lidar. The measurement heights are specific to the Cabauw site, all other parameters are generic.**

| | |
|---|---|
| Laser wavelength | 1560 nm |
| Optical laser power | 1.3 W |
| Ranging | Geometric focusing |
| Horizontal wind retrieval | VAD scan |
| Measuring heights | 11, 20, 39, 80, 140, 200, 252 m agl |
| Scan dwell time | 1 s |
| Height instrument | 1 m |

## 2.2 Description of data and field site

Data from a visibility sensor and from the wind lidar are retrieved for two sites: Cabauw, Netherlands and the UK remote sensing test site in Pershore, UK.

The UK remote sensing test facility at Pershore is located in a flat, rural setting at a former airbase (52.143° N, 2.037° W, Fig. 2a). The Pershore data set covers 24 months, from 1/1/2018 to 1/1/2020. A single visibility sensor of type Campbell Scientific 120/125 was available at Pershore, mounted at 2 m agl on a mast on a meteorological measurement site operated by the UK Met Office, located ~600 m away from the lidar (Fig. 2b). The Campbell Scientific 120/125 uses a 42˚ scatter angle (Cambell Scientific CS125 user manual). The measurement accuracy is ±10% (visibility up to 10 km), ±15% (visibility up to 15 km) and ±20% for visibilities above 15 km.

The Cabauw Experimental Site for Atmospheric Research (CESAR) is located in an extended and flat polder landscape, about 40 km off the coast, 0.7 m below mean sea level (51.971˚ N, 4.927˚ E; Fig. 2a), run by the Royal Netherlands Meteorological Institute (KNMI) and is part of the Ruisdael Observatory. The visibility data used were acquired during a wind measurement campaign described in detail in Knoop et al. (2021) and their measurement is part of a regular observation program (Bosveld, 2020). The coinciding data from the wind lidar and the visibility sensors for Cabauw used here cover about 24 months, from 15/2/2018 to 29/2/2020. The visibility was measured with visiometers of type Biral SWS100 at heights agl of 10 m and 20 m at a meteorological mast called mast B, and at 40 m, 80 m, 140 m and 200 m at mast A (depicted in Fig. 2c). The visiometers measure the forward scattered intensity at 45˚ (Biral SWS100 user manual). The accuracy of the visibilities as reported in the user manual are between ±10% (up to 16 km) and ±20 km (16 km to 30 km). The relevant measurement heights agl of the wind lidar were 11 m, 20 m, 39 m, 80 m, 100 m, 140 m and 200 m (Table 1). The ZX 300 wind lidar was located 293 m away from mast A and 267 m away from mast B (Fig. 2b). Due to the extended probe length of the lidar, height mismatches of 1 m between lidar measurement height and sensor height are expected to be insignificant.

Both visiometer types at Cabauw and Pershore use the same principle of operation, so that the differences between the two types of visiometers are expected to be much smaller than between visiometer and wind lidar. All data from the visibility sensors are time series with visibility in units of metres. As stated above (Eq. 4), the backscatter coefficients from the wind lidars are recorded as time series in units of $1.3 \times 10^{-6}$ m$^{-1}$ sr$^{-1}$. Both backscatter and wind data are measured every second for a given height, but are averaged over 10 minute periods.

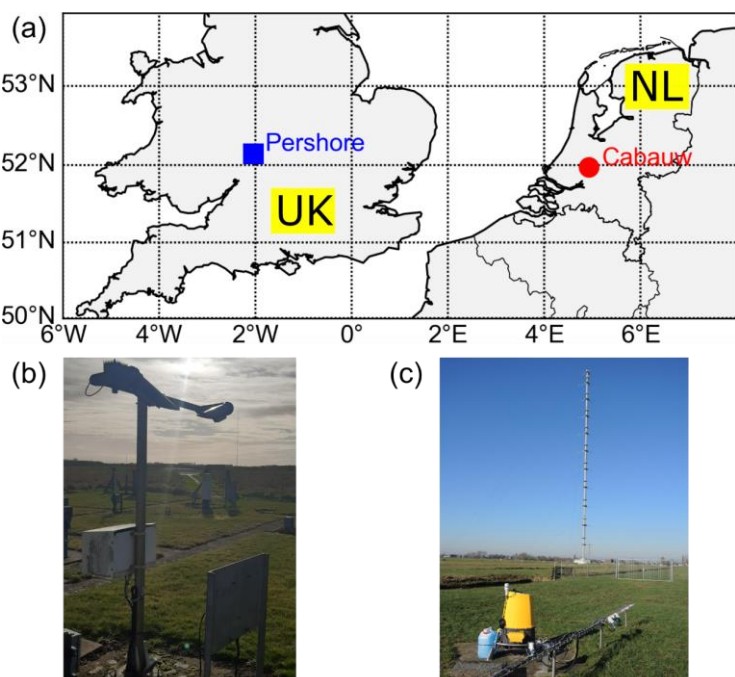

**Figure 2. Location of lidar and visibility sensors (visiometers). (a) Pershore (United Kingdom, UK) and Cabauw (Netherlands, NL). (b) Visiometer at Pershore with meteorological mast in the background where the wind lidar is located. (c) ZX 300 wind lidar at Cabauw with meteorological mast A in the background (From Knoop et al., 2021).**

## 2.3 Retrieval of visibility from the backscatter coefficient

### 2.3.1 Method A: Directly converting backscatter to visibility

The retrieved backscatter coefficients from Eq. (4) are directly related to visibility as follows. The ratio between the extinction and the backscatter coefficient, the extinction-to-backscatter ratio (also called lidar ratio) $S$ is used to estimate the extinction coefficient from the backscatter coefficient as (Doherty et al., 1999)

$$\sigma = \beta S, \tag{6}$$

where $\beta$ is the backscatter coefficient. $S$ is assumed to be constant (Young and Vaughan, 2009). However, $S$ can vary for

common atmospheric aerosols from 1 to about 100 sr, depending on the SD, shape, and chemical composition of the particles (Fernald et al., 1972; Doherty et al., 1999), as will be discussed further below. The empirical power law by Ångström is used to relate the extinction coefficient measured at the lidar wavelength of ~1550 nm to 550 nm (Nebuloni, 2005; Schuster et al., 2006; Shang et al., 2017) as

$$\sigma_0 = \sigma_1 \left(\frac{\lambda_1}{\lambda_0}\right)^{\alpha}, \tag{7}$$

where $\alpha$ is the Ångström exponent, $\lambda_0$ and $\lambda_1$ are the wavelengths corresponding to the extinction coefficients $\sigma_0$ and $\sigma_1$, respectively. Combining Eqs. (1), (6) and (7) yields an approximate MOR of

$$MOR \approx \frac{3}{\beta \, S \left(\frac{\lambda_1}{\lambda_0}\right)^{\alpha}}, \tag{8}$$

which is used to relate backscatter from the lidar to visibility. $\lambda_0$ and $\lambda_1$ are set to 1560 nm (Table 1) and 550 nm, respectively. The two unknowns, $\alpha$ and $S$, depend on the aerosol type, and are a non-unique combination, which can be obtained by

matching the converted backscatter with the visibility from the in situ sensor.

### 2.3.2 Method B: Fitting a transfer function

Coinciding visibility and backscatter data are selected and binned into 120 values of backscatter and 80 values of visibilities. Figure 3 shows a scatter plot of the resulting distribution, representing a 2D-histogram, with the logarithm of the inverse

visibilities plotted against the logarithm of the lidar backscatter.

For the Cabauw visibility data, depending on the availability of both backscatter and visibility at a given height, and depending on the height, there are between ~13000 and ~25000 samples, that is, coinciding visibility/backscatter pairs. There appears to be a secondary mode (Figs. 3a and b), the origin of which is not entirely clear. A reasonable explanation could be a contaminated visibility sensor window beyond the capability of the self-correcting algorithm of the visibility sensor, or it could

be residual cloud contribution. Visibilities greater than 20 km in the visiometer data were set to 20 km during acquisition, which explains the disproportionally high accumulation of values at that visibility. Most of the data is concentrated below 20 km visibility, which therefore is deemed an appropriate upper range. It is obvious from Figure 3a that the data correlate linearly

only for a limited parameter range. Towards lower visibilities, the dependence becomes increasingly nonlinear. Only visibilities of at least 4 km are considered, which helps to select a data range with reasonably linear correlation between

backscatter and inverse visibility and excludes the impact of fog or cloud on the visiometer readings.

For Pershore, data for the same range of visibilities as for Cabauw are selected. Visibility data from a single height only (2 m agl) were available. The nearest lidar data was for a measurement range of 10 m agl, which was only available for a relatively short period, corresponding to only ~9000 coinciding value pairs (samples). Therefore, lidar data from 39 m were chosen, associated with ~28000 coinciding samples (Fig. 3b). The vertical separation is justified further below. There appear

to be two modes: A correlation with relatively flat slope for high visibilities, where data density is highest, and a steeper mode for visibilities below ~20 km.

The curvature in Figures 3a and b, similar to measurements at other locations (Fenn 1966), suggests a linear relationship of visibility and backscatter only over a limited range, as opposed to the relationship between visibility and extinction coefficient (Eq. 1). This implies that the lidar ratio (Eq. 6), is constant only for a limited range of visibility (or

backscatter).

After binning, for each visibility, a threshold $t$ is applied over the spectrum of backscatter values, which is computed as

$$t = \mu + \delta, \tag{9}$$

where $\mu$ is the mean sample density of the backscatter spectrum and $\delta$ is an adjustable parameter. The thresholding is used to

exclude spectral outliers and artefacts (deemed to be unlikely values) and to tighten the distribution of backscatter values at given visibilities (reduce the variance, or spread), corresponding to a forced reduction in samples. After thresholding, the first moment of the backscatter distribution (centroid) is retrieved for each of the 80 visibilities. This corresponds to a maximum likelihood estimation of the backscatter, i.e., the estimation of the expectation value of the distribution of backscatter-visibility sample. The retrieved backscatter values for all 80 visibilities are then used for a linear fit, which also is a measure of linear

correlation between visibility and lidar backscatter. The linear fit is then used as a transfer function (similar to a calibration) to translate a measured backscatter value from the wind lidar to visibility (or vice versa if desired).

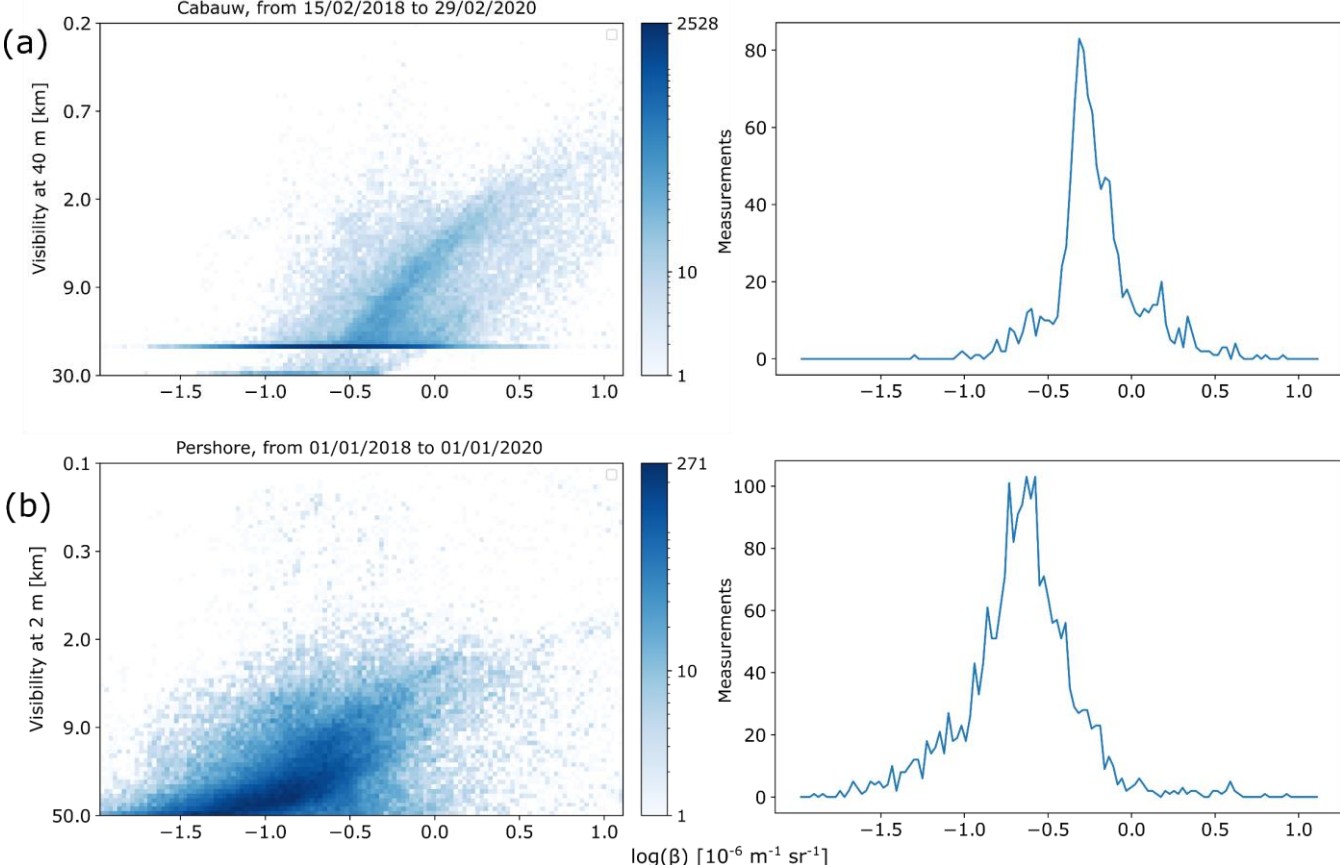

**Figure 3. Overview 2D-histogram scatter plots of inverse visibility versus backscatter (both logarithmic). For better readability, y-tick labels of visibility are shown. The backscatter is logarithmic to facilitate display due to its large dynamic range. (a) Scatter plot for Cabauw, with backscatter distribution at 12 km visibility, indicating a peak at $10^{-0.3} \times 10^{-6}$ m$^{-1}$ sr$^{-1}$. Visibility measured at 40 m agl, backscatter from 39 m agl. (b) Scatter plot for Pershore, with backscatter distribution at 12 km visibility. Visibility measured at 2 m agl, backscatter from 39 m agl.**

 ## 3. Results

### 3.1 Method A

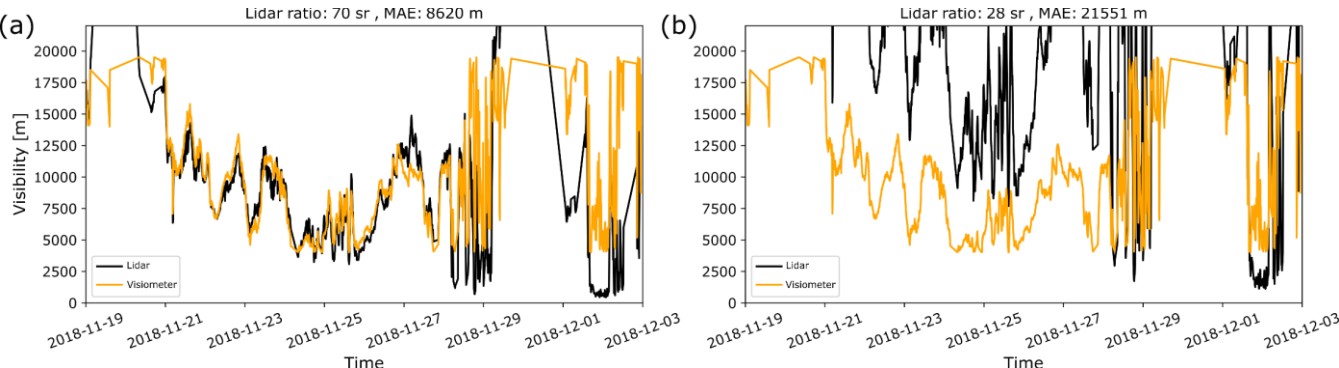

**Figure 4. Linking lidar backscatter from 39 m agl to visibility for Cabauw using Method A. (a) Lidar ratio is 70 sr (continental). (b) Lidar ratio is 28 sr (marine), the Ångström exponent is 2.0 for both. The visibility sensor data are from 40 m agl.**

Equation (8) was used to estimate visibility from wind lidar backscatter for Cabauw from the full visibility range (no selection or filtering was applied, see Fig. 3). The visibility estimate, Eq. (8), is very sensitive to the Ångström exponent $\alpha$, whereas the choice of the lidar ratio $S$ is rather forgiving. The lidar ratio was held constant at 70 sr, representing a continental tropospheric SD (Jäger et al., 1995, Doherty et al., 1999). A typical value for the Ångström exponent has been empirically determined as 1.4 for visibilities between 6 and 20 km (Nebuloni et al., 2005). This value largely overestimated visibilities from the visiometer. Increasing the Ångström exponent to 2.0, associated with a finer, more continental aerosol dominating the backscatter, improved the fit considerably and the result is shown in Fig. 4a. The scatter of the data and the nonlinear dependence between backscatter and visibility (Figs. 3a and b) translates into mismatches between the time series of backscatter and visibility. This does not explain why for some periods the visibilities match well (21/11 to 25/11/2018) and for some they do not (19/11, 29/11 to 3/12/2018). A more marine type lidar ratio of 28 sr (Doherty et al., 1999) associated with a larger aerosol mode size improves the fit somewhat for early December 2018, whilst largely overestimating the visibilities from the visibility sensor overall (Fig. 4b). The mean absolute error (MAE) for the whole period of 24 months of data is 8.6 km for a lidar ratio of 70, and over 21 km for a lidar ratio of 28, suggesting that the dominant aerosol at Cabauw is of continental type. The poor agreement for midday 19/11 and around 30/11 persists (Fig. 4a). This may partly be due to the visiometer data levelling off at 20 km visibility. The backscatter at midday of 19/11 was indeed ~4 times lower than during the morning of 19/11, hence with the same combination of $S = 70$ sr, $\alpha = 2$, thus leading to a fourfold increase in visibility (~58 km) retrieved from the lidar backscatter. The aerosol SD and hence the Ångström exponent and the lidar ratio usually vary over time (Doherty et al., 1999), thus leading to mismatches as in Fig. 4.

As demonstrate above, by varying the Ångström exponent, a coarse fit could be produced, which could then be fine-tuned using the lidar ratio. Of course, this yields a nonunique solution. For example, $S = 28$ sr, $\alpha = 2.6$ yield a similar fit as in Fig. 4a, including MAE. Therefore, assumptions about the lidar ratio and Ångström exponent, would have to be made. These

parameters would need to be measured separately in parallel to lidar backscatter, such as with a nephelometer in combination with a sun photometer (Doherty et al., 1999). Although the Ångström exponent does vary over short periods of time (hours to days), it does so in a confided manner. For certain sites, including Cabauw, Ångström exponents are available from the aerosol optical depth (AOD) product from the Aerosol Robotic Network (AERONET, https://aeronet.gsfc.nasa.gov/new_web/data_description_AOD_V2.html, last accessed 28/07/2022). For a few sites, lidar ratios are measured within the Portable Raman Lidar Network (PollyNet, Baars et al., 2016). For Cabauw, although none of these are available for autumn and winter 2018 from both networks, the Ångström exponent retrieved within AERONET at Cabauw varies usually between approximately 0.1 and 2.0 over the course of any given day and is bounded by these limits over the course of a year. This makes $S = 70$ sr, $\alpha = 2$ more likely than $S = 28$ sr, $\alpha = 2.6$. Figure S1 shows an example where time series of Ångström exponent at Cabauw from the AERONET (O'Neil et al., 2001) were used to improve the agreement with the visiometer especially for lower visibilities, while the agreement for the higher visibilities reduced, since the lidar ratio was assumed constant.

While Method A is an interesting exercise, it is questionable whether it would be practical enough in obtaining a general transfer function between lidar backscatter and visibility. It could be feasible where data from monitoring networks such as AERONET exist, which yields time series of the Ångström exponent and lidar ratio.

## 3.2 Method B

To that end method B was applied to the Cabauw data. A limited visibility range in the 2D-histogram of [4 km, 20 km] (Fig. 3a) was selected in order to increase linearity of the correlation. This was followed by thresholding using $\mu + 1.5$ (Fig. 5a). After the thresholding is applied, the secondary mode has little influence on the fit, due to the relatively few points associated with these backscatter/visibility pairs (Fig. 5b). The resulting histogram reveals a good correlation between visibilities from the visibility sensor and the backscatter values from the wind lidar.

The lidar backscatter coefficient can be quite dynamic. The lidar backscatter coefficients at both sites was observed to fluctuate by up to 5 times within 10 minutes. To assess whether the 10 min averaging window caused any deterioration of the correlation, selected series of backscatter were offset by up to 5 minutes before averaging, with no significant effect on the correlation with the 10-min visibility time series.

The ratio of the number of measurements with a given visibility versus the number of total measurements was computed to assess how frequent a given minimum visibility is. For comparison, this was done using visibility values from the lidar backscatter and visibility values from the visiometer. As a result, using the visibility sensor data only, about 60% of the time visibility is at least 8 km (Fig. 5c). This may be compared with the fraction of measurements for a given visibility as retrieved from the centroid backscatter values, which are in line, but slightly off, especially for the higher visibilities (68%, Fig. 5c). This is caused by the slight nonlinearity in the correlation plot (Fig. 5a).

Method B was applied to the Pershore data set (Fig. 6a). Owing to the spread in backscatter, the R-squared value is significantly lower than for Cabauw (0.80). The difference to Cabauw in intercept and slope of the transfer function (Fig. 6a) indicates that the annual average backscatter is below that of Cabauw for all visibilities considered here. This becomes easily visible when comparing the 1D-histograms (Figs. 5b and 6b). This implies that for the same backscatter, the visibility at Pershore is smaller than at Cabauw. In line with the lower R-squared of the fit (Fig. 6a), the discrepancy is higher between the prevalence of a given visibility for lidar derived versus sensor derived visibilities (Fig. 6c). For example, a minimum visibility of 8 km was measured 62% of the time with the visiometer, but 82% of the time using lidar derived visibilities from the transfer function. The closest agreement is for visibilities around 13 km, which is similar to Cabauw (Fig. 5c).

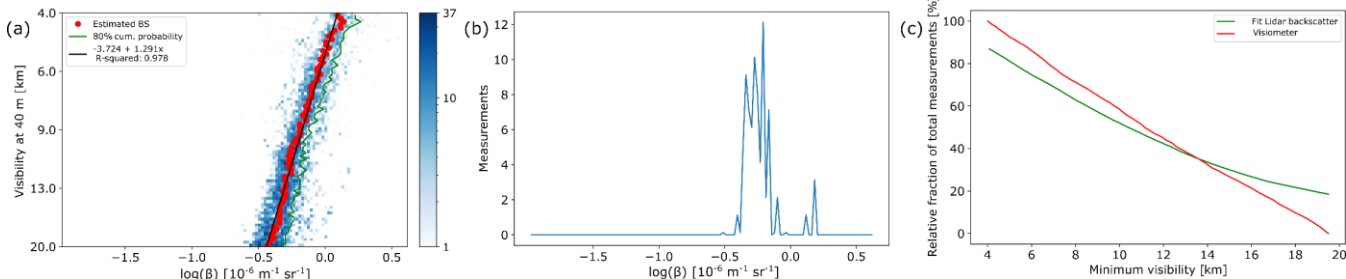

**Figure 5. Visiometer readings versus lidar backscatter, for Pershore. 39 m (2 m) agl for lidar (visiometer). Data covers full range from 01/01/2018 to 01/01/2020. (a) 2D-histogram. Overlain are the centroid backscatter values (red dots), 80% cumulative probability (green) for each visibility and the transfer function (black), to be read as $log(V^{-1}) = -3.724 + 1.291x$, where $V$ is visibility, $x = log(\beta)$, $log$ depicts the decadic logarithm. (b) 1D-slice at visibility of 12 km. (c) Relative fraction of total measurement for a given visibility using the fitted BS values and the visiometer data only.**

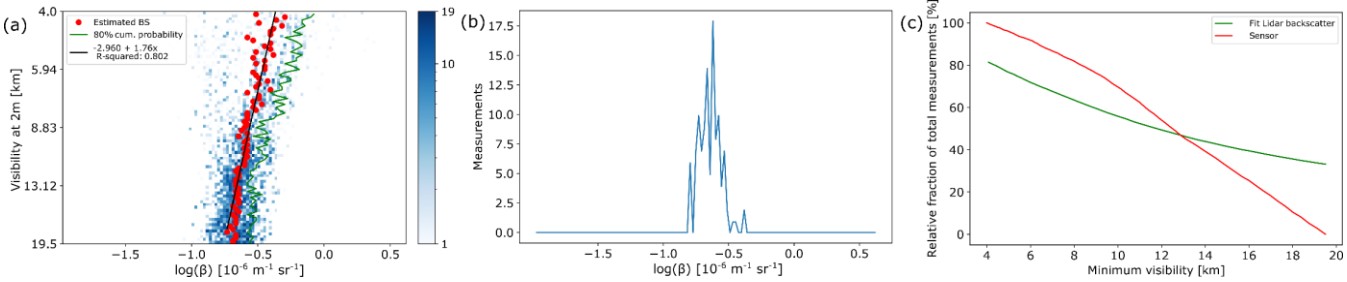

**Figure 6. Visiometer readings versus lidar backscatter, for Cabauw. 39 m (40 m) agl for lidar (visiometer). Data covers full range from 13/07/2018 to 29/02/2020. (a) 2D-histogram. Overlain are the centroid backscatter values (red dots), transfer function (black), 80% cumulative probability (green) for each visibility. (b) 1D-slice at visibility of 12 km. (c) Relative fraction of total measurement for a given visibility using the fitted BS values and the visiometer data only.**

### 3.2.1 Height dependence Cabauw

Only Cabauw data are considered since Pershore data was available for a single height only. While the correlation plots and the transfer functions for different heights of sensor and lidar do not reveal it directly (Fig. 7a to f), the visibilities computed from the fitted transfer functions of these correlations, depending on the given backscatter value, show a possible slight upwards trend with increasing height (Fig. 8). It appears that visibility below ~80 m agl varies only little with height. This is

expected, since, at least in unstable and neutral atmospheric conditions, no significant aerosol stratification at these low heights would be anticipated. It is also in line with other, long-term ZX300 lidar observations from various sites across the globe, suggesting a vertically weakly exponential decrease in lidar signal strength (hence backscatter) that becomes significant above ~200 m agl, corresponding to an increase in visibility. These vertical trends can also be identified in data from the Caliop satellite borne aerosol lidar (Winker et al., 2013). The temporal mean of the visibilities between 4 km and 19.5 km from the
visiometers follow a similar trend (Fig. 8).

To assess the severity of separating visiometer height and lidar probe vertically, lidar backscatter from 39 m was correlated with visibilities from sensor heights not matching the lidar height. At a fixed backscatter value ($0.65 \times 10^{-6}$ m$^{-1}$ sr$^{-1}$), for the correlation of 39 m lidar vs. 200 m visiometer data, the difference in visibility to the visibility from the collocated visiometer/lidar data (lidar at 39 m, visiometer at 40 m agl) is 35% (~3 km), with R-squared dropping to 0.81. Below that
height, the difference is within 9% (800 m), that is, 1%, 9%, 3%, 3% with an R-Squared of 0.96, 0.97, 0.95, 0.89 for 10 m, 20 m, 80 m, 140 m, respectively. This is comparable to the relative change in visibility for heights up to 80 m agl. This indicates that, at least for this case, the visibility sensor and lidar probe height may be separated by few tens of metres, provided a generally well-mixed atmosphere within the layer, in which the sensor and lidar probe are located (as assumed for Pershore). This is also suggested by the time series of the visibility sensors, which, for the heights of 10 m, 20 m, 40 m and 80 m agl
largely correlate. Whilst this can be often be assumed for the continental boundary layer, especially during daytime, when convective mixing takes place, this would less likely to be expected offshore, for instance, due to the presence of advection fog near the sea surface, causing strong vertical gradients in aerosol density and SD.

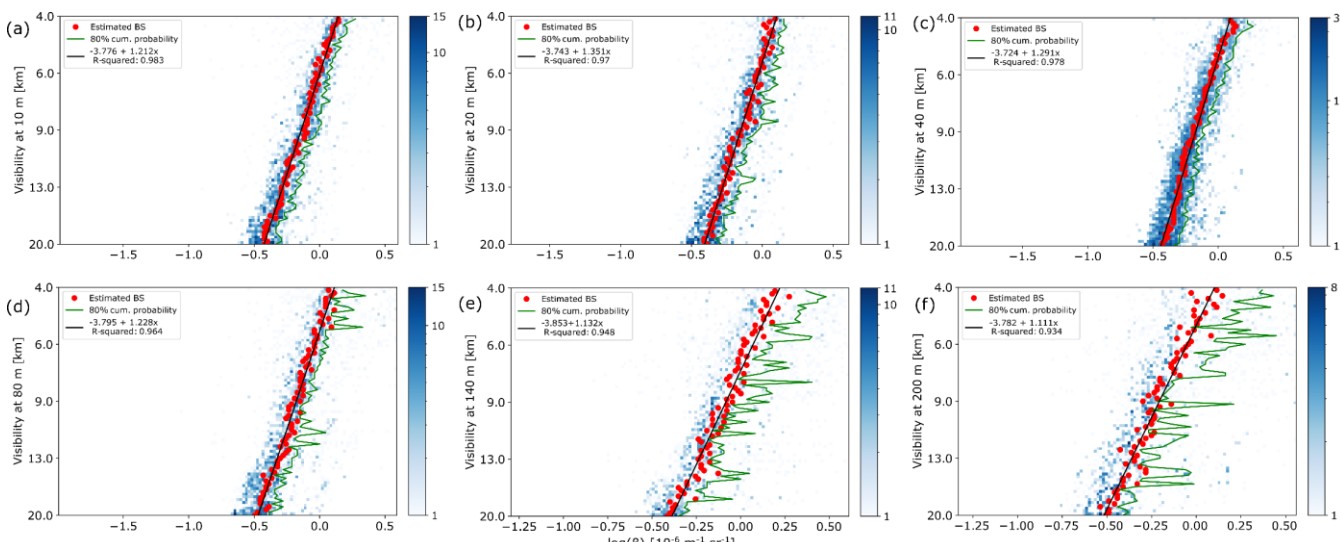

**Figure 7. 2D-histograms using maximum available range of backscatter values for various measurement heights agl. (a) 10 m. (b)**
**20 m. (c) 40 m (d) 80 m. (e) 140 m. (f) 200 m above ground level. Overlain are: Centroid backscatter (red dots), transfer function (black), 80% cumulative probability (green) for each visibility.**

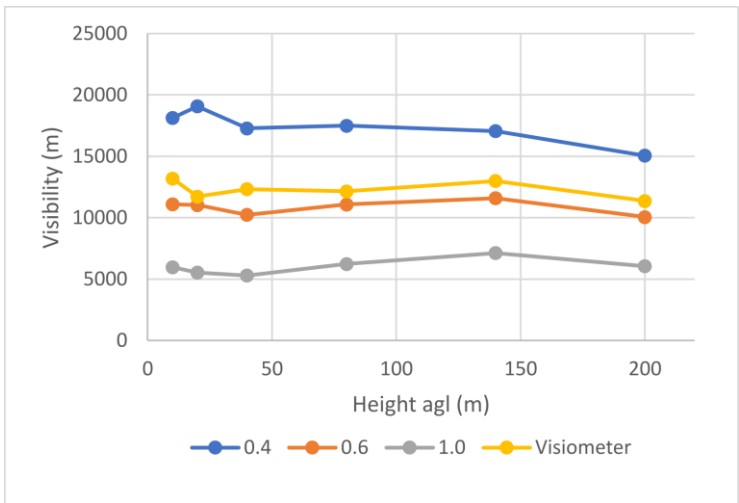

Figure 8. Visibility computed from all six transfer function from Fig. 7 for three different backscatter values (in units of $10^{-6}$ m$^{-1}$ sr$^{-1}$) and temporal mean (15/2/2018 to 29/2/2020) of visiometer readings between visibilities of 4 km and 19.5 km as a function of height.


### 3.2.2 Seasonality

Figure 9 shows 2D-histograms for 40 m agl, which underwent the same processing, as described in Method B, but split by seasons. The threshold has been lowered to $\mu + 1$ to partially compensate for the decrease in sample number which, if below ~4000, may deteriorate the linear correlation. The correlation still deteriorates for summer, due to fewer data points and a larger scattering (only ~2500 data points vs ~8000 for other the seasons), especially for the less frequent lower visibilities. For Cabauw, the backscatter for all visibilities undergoes a strong shift to lower values from spring to summer to then remain relatively unchanged until autumn. Between autumn and winter, the backscatter mainly for the lower visibilities decreases, causing a decrease in slope of the transfer function. A backscatter of $0.5\times10^{-6}$ m$^{-1}$ sr$^{-1}$ ($-0.28\times10^{-6}$ m$^{-1}$ sr$^{-1}$ in Fig. 9a) would give a visibility of ~13 km in spring, ~9 km during summer and winter and 8 km during autumn. For Pershore, the backscatter is distributed differently with respect to visibility, but indicates a decrease from spring to summer and an increase from summer to autumn. A backscatter of $0.5\times10^{-6}$ m$^{-1}$ sr$^{-1}$ corresponds to visibilities of 2500 m, 1800 m, 4400 m, 1300 m (spring, summer, autumn, winter).

Reorganizing the backscatter data into monthly averages shows a clear seasonality for both Cabauw and Pershore (Fig. 10), which the 2D histograms do not reveal directly. As with the 2D-histograms, the monthly averages show a systematic difference in backscatter between the two sites that will be discussed further below. For both sites, the backscatter is highest in the winter and lowest in the summer (Fig. 9). A backscatter minimum around July has been measured with different ZX300 CW wind lidar systems at the ZX site near Ledbury, UK, but also other locations in the Northern Hemisphere (Scott Wyle, ZX Lidars 2022, personal communication). Analysing monthly mean backscatter from Pershore for 9 years (2012 to 2020) resulted in a standard deviation of monthly mean backscatter over the years between 0.02 (July) and 0.09 (March). The z-score varied

between 0.04 and 2.1. Depending on the year and month, the monthly mean backscatter, therefore, differed from the mean over 9 years by 0.04 to 2.1 standard deviations.

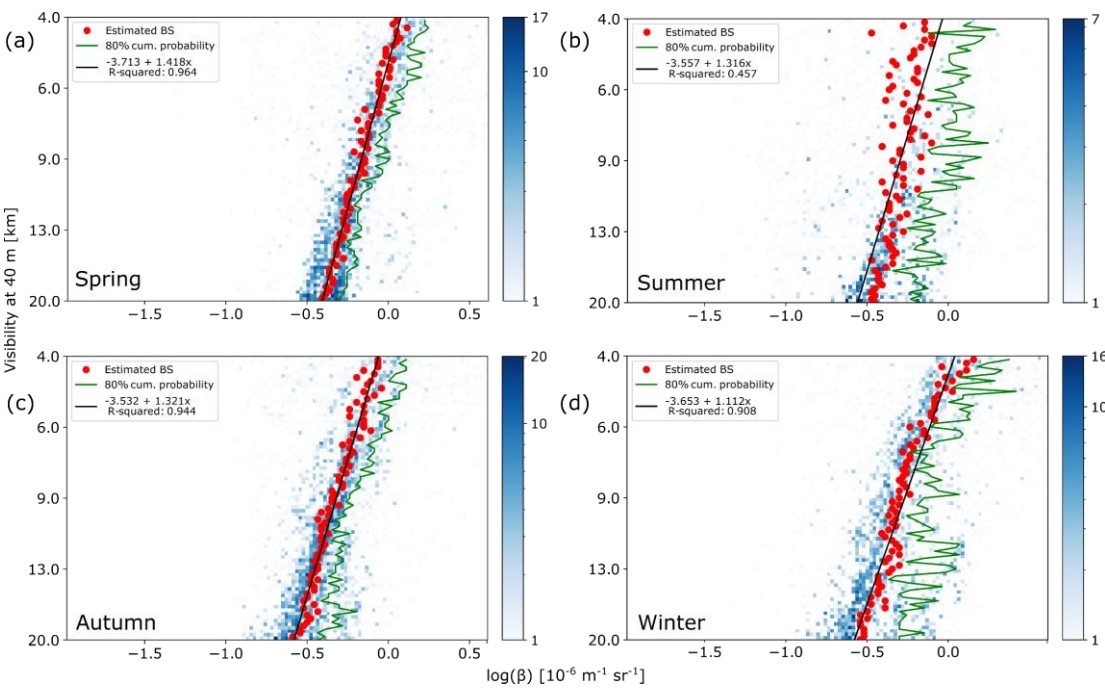

**Figure 9. Cabauw data histograms split by seasons for visiometer height 40 m agl. (a) Spring. (b) Summer. (c) Autumn. (d) Winter.**

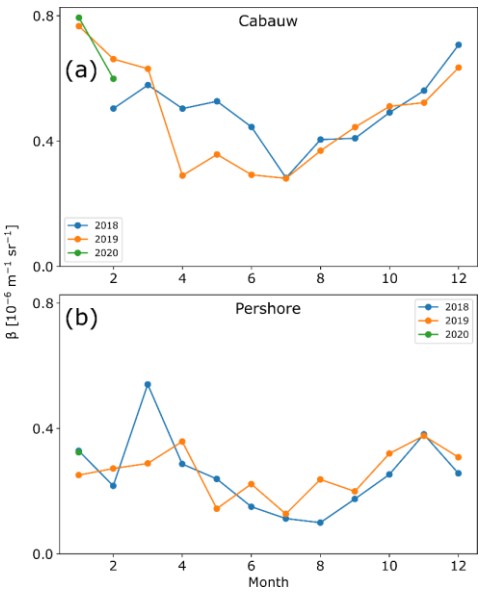

**Figure 10. Monthly mean backscatter 39 m agl. (a) For Cabauw. (b) For Pershore.**

### 3.2.3 Visibility time series

Figure 11 shows a visibility time series for Cabauw derived from lidar backscatter using the fitted transfer function (method B), covering the same period as Fig. 4 (method A). The lidar visibility was derived from lidar backscatter from 39 m agl against visibilities from the sensor at 40 m agl (Fig. 5). At places, the lidar derived visibilities agree well with visibilities from the visiometer (e.g. 21/11 to 26/11/2018), sometimes they largely disagree (e.g. 27/11), leading to an MAE of ~4 km for the whole 24 months covered by the data (an improvement over the MAE for method A, Fig. 4) and 2.1 km for the plotted period (Fig. 11a). The seasonality in the transfer function (Figs. 9 and 10) and the fact that the visiometer data are part of the fitting process, suggest that the closer the data acquisition period used for fitting the transfer function matches the period to predict visibilities, the better the agreement should become. Since the time series in Fig. 11 up to November corresponds to autumn, a reasonable period to choose data from should be autumn. When using data from autumn only to fit the transfer function, the MAE increases slightly from 2.1 km to 2.8 km, which is somewhat unexpected and will be investigated further below. The agreement around 27/11, however, improves (Fig. 11b). The MAE for the whole period of 24 months increased from ~4 km to ~5 km, as one would expect, since autumn only data were used to estimate visibility for all four seasons. When only data acquired in November are used for prediction (two months of data, 11/2018 and 11/2019), the agreement for the period displayed worsens (MAE 2.6 km), while the MAE for the whole 24 months remains at ~5 km (Fig. 11c). A very similar result is obtained, when data from November 2019 only are used to predict visibilities for the period displayed (Fig. 11d).

These tests do not support the hypothesis that the agreement between lidar derived visibilities and visiometer readings improves when the periods for fitting and predicting visibility match more closely. To assess that, other time periods underwent the same procedure with similar outcome. It was found that limiting the data acquisition period practically decreases the variance of the backscatter distribution at a given visibility, i.e., has a similar effect as increasing the threshold (Eq. 9), at the expense of an increased risk of a poorer linear fit, which could explain an increase in MAE. In particular, regardless if spring, summer, autumn or winter was chosen, the overall MAE and the MAE of the period of interest decreased to comparable amounts (4 to 5 km and ~2 km, respectively). Limiting the data period further (or increasing the threshold above a certain level) may decreases the number of data and hence the goodness of the fit, at which point the MAE may increase (as between Figs. 11a and b). For a very large data set (e.g. 10 years of data), however, matching the data period for the fit to that for the prediction could possibly be beneficial to predict more accurate visibilities.

Decreasing the range of visibilities was also tested. It was found that MAE improved, at the cost of a lower dynamic range of visibilities in the time series, since the lidar derived visibility time series covers only visibilities that were used in the

fit of the transfer function. For instance, decreasing the range of visibilities to [7, 15] km led to a MAE of 1.6 km and 2.1 km, for the plotted period and the whole 24 months, respectively.

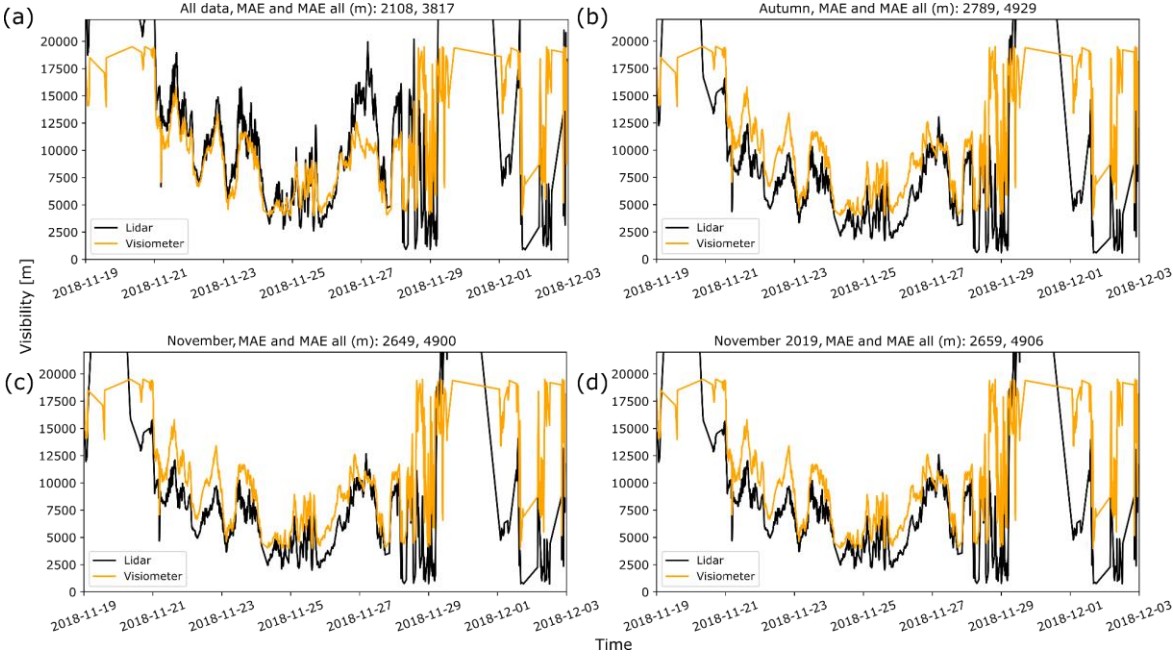

**Figure 11. Lidar derived visibility time series and readings from visiometer for Cabauw. Also shown is the mean absolute error (MAE) between the two time series. MAE is for the plotted period only, MAE all is for whole data period of 24 months. (a) Fit used all data. (b) Fit used autumn data only. (c) Fit used data from Novembers only. (d) Fit used data from November 2019 only. Lidar**
**derived visibilities outside the [4,20] km range used for retrieval have been excluded from the calculation of MAE, but are plotted to illustrate the effect of extrapolating the transfer function outside that range.**

Figure 12 shows the visibility time series for Pershore, derived from lidar backscatter from 39 m agl against visiometer visibilities at 2 m agl (Fig. 6). Compared to Cabauw, the data for Pershore are less tightly clustered around the centroid backscatter values, especially for lower visibilities. Since the fit can approximately be interpreted as the visibility that
corresponds to a given mean backscatter value with the width of the distribution indicating deviation, the larger spread of backscatter values leads to a correspondingly larger MAE between the time series of the lidar derived visibility and sensor visibility (Fig. 12). In other words, a given backscatter value corresponds to a larger range of visibilities. Increasing the threshold reduces the spread of the backscatter values at given visibilities (Figs. 6a and b) and, in fact, it reduces the MAE between the time series (Fig. 12a). An optimum threshold of $\mu + 1.5$ was found (Eq. 9), which avoids deteriorating the linear
fit (R-squared). As the threshold is increased, slope and intercept approach similar values as those for Cabauw. However, since the scale is logarithmic, even small differences lead to significant discrepancies in visibility and backscatter. For a given backscatter, and the visibility ranges regarded here, visibility remains smaller for Pershore than for Cabauw. A reduction in MAE (overall MAE) from 3.8 km (4.8 km) to 3.5 km (4.4 km) is achieved when autumn data are selected (Figs. 12a and b).

MAE remains unchanged for November data and decreases to 3.2 km (4.2 km) if only November 2019 data are used to predict
visibilities.

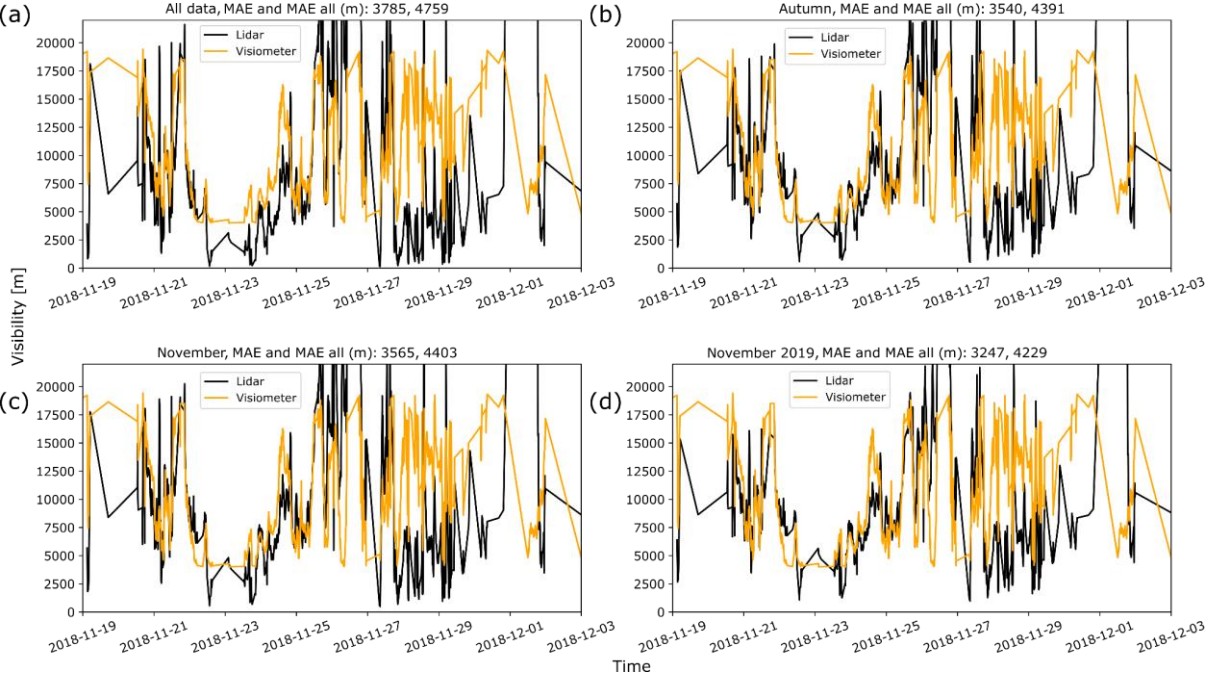

**Figure 12. Lidar derived visibility time series and readings from visiometer for Pershore. Also shown is the mean absolute error (MAE) between the two time series. MAE is for the plotted period only, MAE all is for whole data period of 24 months. (a) Fit used**
**all data. (b) Fit used autumn data only. (c) Fit used data from Novembers only. (d) Fit used data from November 2019 only. Lidar derived visibilities outside the [4,20] km range used for retrieval have been excluded from the calculation of MAE, but are plotted to illustrate the effect of extrapolating the transfer function outside that range.**

## 4 Discussion

The backscatter values at a given visibility form a fairly mono-modal distribution. The transfer function is the linear least square fit of the first moments (centroids) of the distribution of backscatter values at given visibilities. It maps the most common backscatter value to visibility. The linear relationship between the centroids of the logarithmic backscatter and logarithmic inverse visibilities over a limited parameter range is in line with theory (Eq. 1) and previous results (Curcio and Knestrick, 1958; Twomey and Howell, 1965; Nebuloni, 2005). The smaller the range of visibilities considered, the more linear

the relationship between logarithmic visibilities and backscatter, and hence the larger the R-squared value of the transfer function becomes. One would therefore expect that decreasing the visibility range would lead to enhanced agreement between sensor and lidar derived visibilities, i.e., MAE decreases. While this is the case, it has little practical relevance, since the visibilities the transfer function would try to match were not included in the fit of the transfer function.

Assuming that the lidar backscatter probability density is monomodal, thresholding is applied to the distribution to remove artefacts and reduce the spread of the distribution at a given visibility. The transfer function changes after thresholding is applied. For example, for Cabauw the difference in visibility as a result of applying a threshold of $\mu + 0$ versus $\mu + 1$, depending on the backscatter, amounts to a few hundred metres. Thresholding reduces data points and therefore may decrease the "goodness" of the linear fit (R-squared), by increasing the fit error variance (squared error). One would expect this to ultimately cause an increase in MAE between lidar derived visibilities and reference visibilities from a visiometer, since the fitted transfer function explains the data less well. On the other hand, the thresholding decreases the spread of backscatter values at a given visibility, which decreases the MAE, provided the number of data points is sufficient to form a mono-modal distribution (> 1 points around the centroid). The results, therefore, indicate that an optimum threshold exists that maximises R-squared, at the same time minimises MAE with reference measurements from the visiometers. In the present work this optimum has been roughly approached by trial and error, but certainly there is potential for an improved procedure.

The backscatter retrieved from the CW wind lidars undergoes seasonal cycling. Annual mean values of different years and monthly means between different years are reasonably comparable (Fig. 10). However, depending on the year, the result suggests uncertainties up to a few kilometres if a transfer function for a given month was used to predict visibility for the same month of a different year. Since the monthly mean backscatter traces quite well the cycling in gross primary productivity (Fleischer et al., 2015), the observed seasonality could be attributed to aerosol removal processes by leaved vegetation (Wedding et al., 1975). Other possible mechanisms include peaks in primary biological material (Held et al., 2008), a seasonality of condensed water aerosol in the planetary boundary layer and seasonal variations in boundary layer height, which influences dilution of aerosols. Seasonality of atmospheric aerosol extinction, scattering and backscatter coefficients have been measured by others, such as with ground-based in situ measurements and lidar, including the arctic (Schmeisser et al., 2018; Shibata et al., 2018) and Spain (Sicard, et al., 2010; Navas-Guzmán et al., 2013). Mean extinction coefficients over several regions on Earth were measured with the Cloud-Aerosol Lidar with Orthogonal Polarization (CALIOP, Koffi et al., 2012). Consistent with the present finding, the mean extinction coefficient over Western Europe had a minimum in the summer. Schmeisser et al. (2018) found that in some locations around the Arctic Circle, maximum aerosol scattering occurred during spring and winter, whilst in other locations maximum extinction was measured during the summer months. This advocates as likely drivers of seasonality of aerosol backscatter regionally different mechanisms as well as the transport of aerosols into the region of measurement, such as from man-made sources (e.g., sulphates and soot particles, Shibata et al., 2018) or natural sources, including Sahara dust (Sicard et al., 2010; Navas-Guzmán et al., 2013) and sea salt (Koffi et al., 2012).

Although both sites have the same qualitative seasonality in backscatter, when sorted by visibility in a 2D histogram (Figs. 5 and 6), the trends are dissimilar, indicating that a given backscatter does not imply the same visibility at a different location. In fact, the backscatter at Pershore is systematically lower than at Cabauw. As mentioned above, differences between the visiometers should be insignificant. There is a contribution from the transmitted optical power of the wind lidars at the two sites. The transmitted power between units may vary by ~6%. Adding differences in receiver sensitivities between the two, the maximum expected difference in backscatter is of the order of 10%. As the observed differences are far greater than that, they

are in all likelihood related to the local aerosol properties, more specifically, to different dominant aerosol types. For instance, a visibility of 12 km corresponds to a backscatter of $0.61\times10^{-6}$ m$^{-1}$ sr$^{-1}$ at Cabauw and $0.26\times10^{-6}$ m$^{-1}$ sr$^{-1}$ at Pershore. Given that on a scale of minutes, the lidar backscatter value at a given site may easily vary by several factors, this difference is small. However, as it represents an average over many minutes, it is significant. Averaged on a monthly basis, the backscatter values at Cabauw are systematically greater than those for Pershore for each month of the year (Figs. 5 and 6). During the summer months, the backscatter at Cabauw is about three times larger, during winter about two times larger than at Pershore (Figs. 9 and 10), suggesting differences in local atmospheric backscattering characteristics that may change with season. Interestingly, for Pershore, the backscatter values scatter considerably more around the centroid backscatter than for Cabauw.

These results are coherent with previous findings that reported a strong dependence of backscatter to the governing aerosol mode (Twomey and Howell, 1964; Vogt 1968), as stated in the introduction of this paper. Although the extinction-to-backscatter ratio (lidar ratio) is often assumed to be constant (including method A), this assumption is strictly only a fair approximation for homogenous scattering, such as Rayleigh scattering. As the size of the scattering particle increases, Rayleigh scattering is replaced by the Mie scattering model, which, assuming spherical aerosols, is predominant for the present study. Under Mie scattering, the assumption of isotropic scattering breaks down and the angular distribution of the scattered electromagnetic field develops a directivity, i.e., an imbalance in the light intensities between forward scattering angles (measured by visiometers) and the backscatter angle of $\pi$ (measured by the wind lidar), which varies with size parameter $\alpha = 2\pi r/\lambda$ and therefore with particle radius $r$ and wavelength $\lambda$ (Vogt 1968, Shang et al., 2017). As the lidar ratio of a spherical particle changes with particle size, backscatter may vary strongly as particle type and size change, even along the beam, while the extinction coefficient varies only little (Tworney and Howell, 1965). The visiometers measure forward scattering, and hence aerosol extinction, at angles with minimum sensitivity to changes in Mie-scattering intensity upon change of particle type and size.

Since extinction and backscatter coefficients are quantities integrated over SD and size parameter, the fluctuation of the lidar ratio (and hence backscatter) with particle size is smoothed. Real world aerosol SD are far from being homogeneous and a smoothing effect can be expected. This also implies that the use of polychromatic light yields a backscatter intensity less dependent on the aerosol SD than the highly monochromatic light of a coherent wind lidar. The result by Tworney and Howell (1965) suggests that, due to this smoothing effect, the spread of backscatter for a given visibility (Figs. 5, 6, 7, and 9) is up to twice as high as it would be for a measurement with polychromatic light. The magnitude of the spread is likely also a function of the variety of aerosol types and hence SD present over the data acquisition period, which increases nonuniqueness of the relation between visibility and backscatter (Fenn 1966). This suggests the possibility that relative to Cabauw, Pershore experienced a higher variety in aerosol type and SD.

The systematic offset between two sites too is likely caused by a different local aerosol SD to which the backscatter is more sensitive than forward scatter at the angular ranges used in the visibility sensors. It appears likely that different aerosol types (or SDs) may give a similar forward scattering intensity, hence similar visibility sensor reading, but different backscatter values. Different sites are associated with a different predominant (mean) aerosol SD, which, therefore, qualitatively could

explain not only the variance (spread) of the lidar backscatter value for a given visibility (e.g. Fig. 6a), but also the offset by a factor of 2 to 3 between the mean backscatter of the two sites (Fig. 10).

Since aerosol density affects backscatter, a mean aerosol number density systematically higher or lower throughout the year would certainly contribute to the observed offset.

         As the difference in transfer function is in all likelihood related to differences in the predominant, local aerosol SD and/or particle number density, this indicates that after calibrating the backscatter measured by the CW wind lidar (method B) using a visiometer, the lidar could be used to measure visibility. The same applies to method A. Since the predominant aerosol SD in the planetary boundary layer is a function of location and the transfer function has been found to vary between different

locations, the transfer function is probably not generalizable, but site specific. Even if the transfer function was similar for aerosol SD typical to a certain setting (e.g. marine, near coastal), it would not necessarily be transferrable to a different location, since the local aerosol SD could become atypical, for instance, if the location is near a heavy polluter, such as a dense urban area (Curcio and Knestrick, 1958; Twomey and Howell, 1965).

         The more it is known about the dominant aerosol probed by the lidar the more the transfer function could potentially

be applied to different locations with similar average aerosol SDs. It is therefore desirable to gain more information on the predominant aerosol type at Pershore and Cabauw. This would require a different set of instruments, which is beyond the scope of this work. But informed estimates can be made nonetheless. Pershore is located in a rural area. During the predominant south-westerly and westerly winds, air coming from the Atlantic Ocean passes a strip of land about 300 km wide before it reaches Pershore. This area lacks major industry hubs or urban areas that would concentrate sea or road traffic and act as

constant and strong polluters. The dominant aerosol SD is thus likely a rural one, which might be disturbed by road traffic. Furthermore, a landfill, located ~800 m to the southwest of the lidar location, could also produce enhancements in Diesel aerosol from the operated machinery and aerosols common to landfills (Nair 2021). This was indicated by a pronounced difference in the transfer function fitted on weekend and weekday data. Less road traffic on the weekend could be confirmed by observation.

During the predominant south-westerly winds, the aerosol type and number density around Cabauw is expected to be heavily influenced by road traffic aerosol from the Rotterdam suburban area, including the sea port (Karl et al., 2016) and remnants of combustion aerosol from the southeast of England, notably the London area (Fig. 2a). The strong concentration of aerosols from both road and sea traffic and industrial air pollution downwind of Rotterdam and Rotterdam harbour may explain the larger average backscatter at Cabauw (Fig. 10) and the corresponding difference in the transfer functions between

the two sites (Figs. 5a and 6b). The difference in slope of the transfer functions is likely dominated by the difference in the lidar ratio, i.e., due to different dominating aerosol type(s) at the two sites.

         Splitting the data into day and night may also yield hints as to which extent man-made aerosols are dominating. However, other diurnal mechanisms, such as boundary layer mixing processes (Stanier et al., 2004), most certainly will affect aerosol type and number density, and hence lidar backscatter. Interestingly, for Cabauw (Figs. 13a and b) the centroid

backscatter increases during the night (18:00 to 6:00) for both 19 km visibility (by ~45%, from $0.28\times10^{-6}$ m$^{-1}$ sr$^{-1}$ to $0.41\times10^{-}$

6 m⁻¹ sr⁻¹) and 5000 m visibility (by ~21%, from $0.86 \times 10^{-6}$ m⁻¹ sr⁻¹ to $1.05 \times 10^{-6}$ m⁻¹ sr⁻¹). A backscatter of $0.65 \times 10^{-6}$ m⁻¹ sr⁻¹ (-$0.19 \times 10^{-6}$ m⁻¹ sr⁻¹ in Fig. 13a) would give a visibility of ~7.1 km for the day fit and ~9.7 km if considering night time data only. For Pershore (Figs. 13c and d), no significant change in backscatter was found during night time for 19 km visibility, but a pronounced decrease for 5 km visibility of 20% (from $0.43 \times 10^{-6}$ m⁻¹ sr⁻¹ to $0.34 \times 10^{-6}$ m⁻¹ sr⁻¹). Though not adequate to

indicate man-made aerosol, splitting the data into day and night is insightful as it further demonstrates the sensitivity of linking visibility to lidar backscatter. For practical application that target daylight visibility, it may be advisable to exclude night-time data before fitting the transfer function.

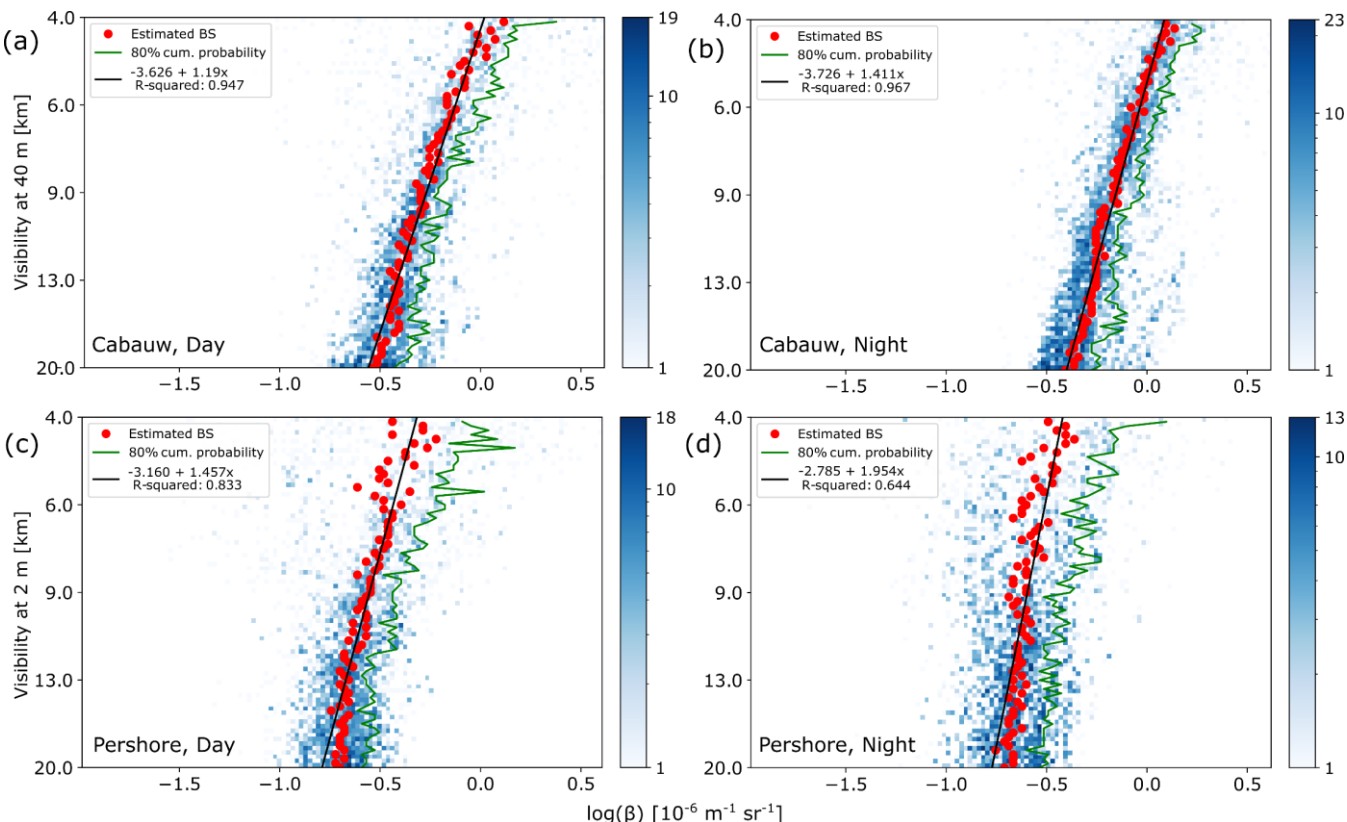

**Figure 13. Comparison of 2D-histograms between day and night time at 39 m lidar probe height (a) Cabauw day. (b) Cabauw, night. (c) Pershore, day. (d) Pershore, night. To account for the reduced sample size, the threshold was reduced to $\mu + 1$.**

In the Mie-scattering regime, which assumes spherical particles, the polarisation of light is preserved. The polarisation transmitted by the wind lidar is random and the polarisation of transmitted and received photons is assumed to be equal. Depolarization effects by aerosols could, however, influence backscatter intensity. The technique used by the visiometers, on

the other hand, is polarization insensitive. Two aerosol types, spherical (e.g. water droplets) and non-spherical (e.g. sea salt aerosols), may thus cause similar scattering intensities, but due to a change in polarization (depolarisation) caused by the non-

sperical aerosol, the backscatter as detected by the wind lidar may differ, which will contribute to a site-specific transfer function.

For Cabauw, lidar backscatter derived visibility was found to be weakly height dependent (Fig. 8), in line with the
observation that under cloud free conditions backscatter from CW-wind lidar usually tends to slightly decrease with height in the lower part of the planetary boundary layer. The coherence of vertical trends of visibility as seen between wind lidar and the visiometer and lidar is encouraging in that it suggests that once calibrated, a CW wind lidar may indeed be useful in generating profiles of SOR in situations where visiometers at heights above tens of m agl are not available or feasible.

**Conclusions**

Backscatter data from two CW-wind lidar systems, at Cabauw (Netherlands) and Pershore (UK), have been analysed with the aim to retrieve meteorological optical range (visibility).

Directly relating backscatter to visibilities was found practical only in cases where additional input data to the backscatter are available, that is, extinction-to-backscatter ratio (lidar ratio) and Ångström exponent.
Calibrating lidar backscatter coefficients with co-measured visibilities from visiometers and fitting a linear transfer function to points of maximum likelihood backscatter was found to be more viable, since, once the backscatter is calibrated, it would not rely on secondary measurements acquired in parallel. For larger ranges of visibility and backscatter coefficients, the correlation was found to be less linear. The method deems, therefore, practical only over a limited parameter range. This implies that the lidar ratio is constant over a limited range of backscatter values only. In addition, the results indicate a spread
of backscatter values for a given visibility, with the spread being dependent on the location. The spread likely corresponds to a nonunique relation between visibility and backscatter.

For a given location and backscatter coefficient, both the nonlinearity and the nonuniquenes are linked to the contribution of a variety of aerosol types and size distributions over the data acquisition period (Fenn 1966). For a given time, but separate locations, differences in local dominant aerosol type lead to differences in lidar ratio and therefore a site dependent
transfer function. In other words, two different aerosol types may give a similar forward scattering intensity, hence similar visibility sensor reading, but a different backscatter and thus a different wind lidar-derived visibility. Therefore, backscatter measurements from CW wind lidar are only representative and repeatable for environments with similar aerosol SDs. This is in line with previous findings (e.g. Curcio and Knestrick, 1958; Fenn, 1966). Both nonlinearity and nonuniqueness are independent of the setup used to measure backscatter (e.g., CW lidar, pulsed lidar, flash light, Curcio and Knestrick, 1958;
Doherty et al., 1999; Werner et al., 2005, p. 172).

The result suggests that backscatter from CW-wind lidar is useful to infer visibility, but it needs to be calibrated against a visibility sensor in an atmosphere similar (similar mean aerosol type) to the one of its intended use, ideally over the course of a year to capture seasonal variation. For the 2-year data set used here, selecting a subset of the data (season, month

etc.) did not improve the accuracy of the transfer function, i.e., accuracy of predicting visibility for the corresponding subset of the same year or a different year. Data sets acquired over more than two years may improve the accuracy of the transfer function.

Going forward, it might be useful to acquire transfer functions at more sites globally and categorize them into sites with similar predominant mean aerosol size distributions. Obtaining visibility data from more sites is desirable to test how site specific the transfer function is and how comparable it is between similar environmental settings.

As far as the two sites assessed in this study are concerned, even after calibrating the lidar backscatter with an in-situ visibility sensor at the site of intended use, the expected accuracy in terms of mean absolute error is over a kilometre. The method would thus deem suitable for safety uncritical applications, such as industrial (e.g. visibility of wind turbines, oil rigs from the shore etc.) or in scientific research. A possible application could include the statistical estimation of the frequency of wind turbines visibility. This could especially be interesting for offshore sites, where, for economic reasons, the distance between the windfarm and the shore has to be minimized. The visibility derived from backscatter could be a modelling input parameter amongst other input parameters, such as solar angle, object colour, Earth's curvature, or cloud cover. Since the backscatter depends critically on the aerosol SD, this could potentially open up applications where sensitivity to aerosol chemistry is desired, such as pollution monitoring or detecting changes in particular matter properties during passive and eruptive degassing phases of volcanoes, which are linked to physiochemical processes inside the volcanoes' plumbing systems.

**Data availability**

Cabauw tower and surface data sets are available as: Meteo profiles - validated tower profiles of wind, dew point, temperature and visibility at 10 minute interval from https://dataplatform.knmi.nl/dataset/cesar-tower-meteo-lb1-t10-v1-2 and Meteo surface - validated observations of common atmospheric variables at 10 minute interval from https://dataplatform.knmi.nl/dataset/cesar-surface-meteo-lb1-t10-v1-0. Pershore visibility data are available at https://zenodo.org/record/6325902#.YiDrNejP2Uk (Queißer et al., 2022). Backscatter data are available from the first and second author on request.

**Author contributions**

SK was responsible for the wind lidar measurement campaign at Cabauw. MQ and MH performed the data analysis and wrote the draft. All authors worked on the refinement of the manuscript.

**Competing interests**

The authors declare that they have no conflict of interest.

**Acknowledgements**

We thank Matt Smith and Scott Wylie (both ZX Lidars) for the inspiration for this paper and helpful discussions.

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
