# Peer review of "Atmospheric visibility inferred from continuous-wave Doppler wind lidar"

_Atmospheric Measurement Techniques, 2022_

## Author Comment (AC1)

Response to reviewer #1

Here, visibility is estimated from a continuous-wave (CW) Doppler lidar and compared with visibility sensors at two locations: Cabauw, Netherlands and Pershore, United Kingdom. Retrieving visibility or aerosol backscatter from CW Doppler lidars would enable further studies with a widely-spread instrument type, which is currently not utilised to retrieve aerosol-related parameters. Therefore, I consider this manuscript within the scope of AMT.

Response:

The authors would like to thank the reviewer for the time and effort taken to review this manuscript. It is very much appreciated.

We authors would like to point out that we believe there is a slight misunderstanding with regards to the validity of the method used, which negatively biased the review criteria mark. It seems that the reviewer is from the ceilometer/backscatter lidar community, which explains the main concern on range corrections etc.. We hope to have made it clear below, that the method to retrieve the backscatter is different to pulsed lidar methods, but it is by no means less valid. It is absolutely valid and well proven. In fact, cw wind lidar has become standard in wind resource assessment (IEC 61400-12-1:2017) and the backscatter retrieval, which is outline in all detail in our manuscript, is part of the theoretical basis of that standard.

We furthermore made improvements to readability of the manuscript by changing the scaling of backscatter value in the graphs as requested by the reviewer and by improving the structure of the discussion and result sections.
Changes made to the manuscript are highlighted in yellow.

Unfortunately, agreement between the CW Doppler lidar and visibility sensor is not very good, and it is questionable how useful this method would be. What is missing, is a more detailed analysis into the reasons for the observed discrepancy.

Response:

We would like to stress that the reviewer's comment "agreement between the CW Doppler lidar and visibility sensor is not very good" would apply only to Pershore, not Cabauw, where the agreement is good as is shown in the article.

The paper describes if and how a readily available data product acquired in various places around the world may be used as is to infer visibility. It does this by discussing two practical cases. The usefulness of the method is extensively discussed in the paper.

Major comments

I am not convinced that the observed differences are due to aerosol properties for the most part. Especially, since the backscatter coefficient retrieved from the CW Doppler lidar is not corrected for range, focus or attenuation. For instance, attenuated backscatter from ceilometers is considered to require calibration before use (e.g. Kotthaus et al., 2016; Hopkin et al., 2019). Also pulsed Doppler lidars require substantial post-processing for aerosol parameter retrievals (e.g. Vakkari et al., 2019; Pentikäinen et al., 2020). Please compare attenuated backscatter from CW Doppler lidar with attenuated backscatter from a reference instrument (e.g. ceilometer with proper post-processing).

Response:

Concerning the comparison of the backscatter retrieval with other methods:

There are fundamental differences in the approaches. Kotthaus et al. use pulsed lidar, we use CW lidar. One can see this when comparing Eq. 9 in Kotthaus et al. (2016), to Eq. 2 in our paper. Equation 2 in our manuscript, $P_S = \pi P_T(\pi)\lambda$, is the lidar equation for a CW coherent lidar with geometrical focus, which has no r^2 dependence that would have to be corrected. The backscatter coefficient retrieved from the CW Wind lidar is proportional to the integrated heterodyne power spectral density of the wind signal, i.e. the signal power, see Eq. 4, integrated over the probe volume.

Pulsed lidars (basically all in your citations below) usually operate in a collimated or weakly focused regime, otherwise they would not measure range resolved profiles of backscatter. The wind lidar signal power we measure is independent of range, since we focus the beam and the lidar thus operates in the backpropagated local oscillator (BPLO) scheme (see cited article in our manuscript: Harris, M., Constant, G., and Ward, C.: Continuous-wave bistatic laser Doppler wind sensor). This means return power is independent of range, as noted in the manuscript, line 139. Therefore, no geometrical correction (r^2) is applicable here. Note that the BPLO scheme is only applicable for relatively short ranges, depending on the telescope aperture a few hundred metres, which is the case here.

In pulsed lidars, including Pentikäinen et al. and including ceilometers, this is different. The backscatter can be associated with a range bin. In our case, on the other hand, we integrate over a probe volume of length Rayleigh length, i.e. over an extended range. Our backscatter coefficient corresponds to the distance the lidar is focused at and is integrated over a Lorentzian-shaped curve (i.e. the Rayleigh range of the focus).

Moreover, the cw wind lidar used at Cabauw and Pershore are monostatic, whereas ceilometers/pulsed lidars are bistatic. Eq. 2 has no overlap function. That means the cw wind lidar does not have the same issues like most ceilometers, which are bi-static. Within the range measured in our paper (0 to few hundred meters), pulsed lidars have 0 or little signal due to the limited overlap of

transmitter and receiver field of view. So even though your suggestion to do a direct comparison on the backscatter to understand its behavior, and then go to visibility, makes sense, in practice it will probably not work.

Regarding data processing, data with extremely low SNR (CNR in our case) are discarded. The coherent detection (we use) is literally immune to solar irradiance. Instrumental drifts are not applicable since the system uses a common mode setup, which means any drifts, such as due to thermal expansion of optical paths, cancel out.  All other parameters in Eq. 4 are constants (output power, wavelength etc.).

The calibration you mention, is discussed in detail in our manuscript: Without secondary input (Angstroem exponent, lidar ratio), method A is not very practical and without calibration with a visiometer method B is not useful. This is one main conclusion of our paper.

We hope this has made it clear that the backscatter retrieved, though not directly comparable to ceilometers, is valid.

It has been established a while ago (e.g. Curcio and Knestrick 1958, Fenn, 1966; see References in manuscript) that there exists a non-unique (in other words site dependent) relationship between backscatter and visibility. This paper is basically an experimental verification of this using Doppler wind lidar. This has nothing to do with the correction you mention, since it does not apply to the backscatter measurement technique used here, as explained above.

There are site dependent differences in the relationship between visibility and backscatter for the two campaigns, Pershore and Cabauw. In addition, for Cabauw we see a good correlation, for Pershore we see a poor correlation, but this is a different issue. Both can only be explained by the site specific difference as I will explain as follows. Both campaigns used the same type of lidar hence the same way of retrieving the backscatter. It is already explained in the manuscript that the differences are not explicable by instrumental differences. That means, backscatter from other types of instruments (e.g. ceilometers) would have resulted in the same differences between the two sites. The point of the paper is to show that there are site specific differences. Backscatter is caused by aerosols and only aerosols. Therefore, under the reasonable assumption that instrumental differences are negligible, site specific differences can only mean that differences are due to aerosol properties.

Retrieval algorithms of backscatter do not matter in this case. It is just basic physics. Even if we were to point a torch in the sky and measure the backscattered power with a cheap silicon photodiode hooked to a voltmeter, the received power would differ between the two sites, even if visibility was exactly the same at both sites and the setup and procedure were exactly the same at both sites.

We have added the following to the methods to ensure the difference to ceilometers is understood:

We changed on l 141

Note that Eq. (2) contains no dependence on either the focus range or the system aperture size

To

==Note the difference to pulsed lidars and ceilometers. Equation (2) contains no dependence on either the focus range or the system aperture size. Moreover, the system is monostatic. That means, the typical measurement range of a CW lidar is 0 to few hundred meters with full overlap of transmitted==

and received beam, which is different to pulsed, bistatic lidars, which have limited overlap in the near field.

Lines 218-220 "Towards lower visibilities, the dependence becomes increasingly nonlinear. Only visibilities of at least 4 km are considered, which helps to select a data range with reasonably linear correlation between backscatter and inverse visibility and excludes the impact of fog or cloud on the visiometer readings." In my opinion, the low visibility end of the spectrum is even more interesting than > 4km range (e.g. fog detection). Please include <4km visibility in the analysis.

Apart from substantial reasons like fog, as mentioned in the manuscript, one main interest for the reader and hence motivation of the paper was turbine visibility from the coastline, which are usually further than 4 km away from the coast (more like 12 km. So the range 4 to 20 encompasses these. In addition, we preferred to include a data range as long as wide as possible, so 0 to 4 km is smaller than 4 to 20 km. Also, there are more readings for 4 to 20 km which makes it easier to fit. So we prefer to stick to the wider range. Less than 4 km would make sense for safety critical applications, which, as concluded in the paper, the method is not suitable for.

Specific comments

83-85 "The backscatter coefficient has a higher sensitivity to the size of the aerosols along the beam path and hence to the aerosol size distribution (SD) than the extinction coefficient." Please provide reference.

Response: Reference (Twomey and Howell, 1965) has been added.

103-105 "Due to the longer wavelength (~1550 nm) of most CW wind lidars compared with visible backscatter lidars described above, at normal working ranges (up to 300 m), the return signal is not sensitive to atmospheric extinction, but is practically governed by the backscatter coefficient only." If visibility is low, I'd expect extinction to substantial. And for many applications

low visibility is the interesting part. Can you indicate a visibility range when extinction can be ignored?

Response:

Here is a rough estimation: Assuming a common visibility of 10 km (slightly hazy), extinction at 550 nm would be 3/1e4 m-1. Using Ångström's law, at 1560 nm (wind lidar) that would be ~ 5e-5 m-1, so maximum extinction (i.e. at range 300 m) would be only exp(-2 x 300 m x 5e-5 m-1) = 0.97. I have added this example in the manuscript. Fog would lead to high extinction due to strong scattering. The signal would stem from the first 10 m or so, regardless of where the lidar would be focused, so the lidar signal would still not be much affected by extinction, as the measurement would be just like reflecting off (a diffuse) mirror

A pulsed lidar, on the other hand, would, due to the range gating, still yield a range resolved backscatter profile with very large backscatter (and attenuation) along the whole height of the fog layer.

The paragraph now reads:

As opposed to pulsed aerosol lidar described above, CW wind lidar has a lower measurement range. In addition, CW wind lidar operates in the short wave infrared region close to 1550 nm, which is a factor of 1.5 to 3 longer than for typical aerosol lidar and ceilometer systems (Werner et al., 2005; Gasteiger et al., 2011; Navas-Guzmán et al., 2013; Shibata et al., 2018). Compared with pulsed aerosol lidar, at normal working ranges (up to 300 m), the return signal of a CW wind lidar is thus not sensitive to atmospheric extinction, but is practically governed by the backscatter coefficient only. This is illustrated in the following example. Assuming a common visibility of 10 km (slightly hazy), the maximum extinction (i.e. at range 300 m) would only be $e^{-600\,m\,5\times10^{-5}m^{-1}} = 0.97$. This leaves the backscatter coefficient (henceforth termed backscatter) as the most obvious proxy of visibility of a CW wind lidar.

Added Reference:

Gasteiger, J., Groß, S., Freudenthaler, V., and Wiegner, M.: Volcanic ash from Iceland over Munich: mass concentration retrieved from ground-based remote sensing measurements, Atmos. Chem. Phys., 11, 2209–2223, https://doi.org/10.5194/acp-11-2209-2011, 2011.

134-135 How good is the cloud removal algorithm? Has it been compared with a ceilometer for instance?

Response:

No it has not been compared with ceilometer. The wind lidar itself detects cloud by measuring their speed, if there is cloud signal. Clouds can be identified in the wind lidar raw spectra and hence removed. Hundreds of wind lidars have been verified over a decade (hence under many different atmospheric conditions) and have shown very good agreement to cup-derived wind speeds. This would not be the case if the algorithm was not efficient. Of course, as with every algorithm, it is not perfect and there can still be outliers.

We have added a bit more information by rephrasing the sentence on l138 to:

A cloud removal algorithm is used to correct for this effect by removing Doppler signal biased by cloud returns, which involves a measurement at an additional higher altitude.

137 Please define "pi".

 I assume you mean β(π). It depict the backscatter at 180 degrees. It could also just write β. But that notation is used elsewhere in the literature, including Harris et al., 2001 which is cited at l 136. We changed it to just β. It depends on only on a single range, the focus range of the lidar (see discussion above about range resolution of CW wind lidar)

We  also added to l138:

(…) and $\beta$ is the atmospheric backscatter coefficient at the focus range

180-181 "As stated above, the backscatter coefficients from the wind lidars are time series in units of 1.3×10-6 m-1 sr-1." Please give backscatter in units of [m-1 sr-1] throughout the paper. Scaling by 1.3 x 10^6 makes it hard to follow the results.

Response:

It already is in units of m-1 s-r-1, the numbers are just divided by that factor. The data product provided by the wind lidar is in units of 1.3×10-6 m-1 sr-1, the reason is given in lines 145 to 154.

To emphasise, this we changed the following phrase from:

As stated above, the backscatter coefficients from the wind lidars are time series in units of $1.3 \times 10^{-6}$ $m^{-1}$ $sr^{-1}$

to

As stated above (Eq. 4), the backscatter coefficients from the wind lidars are recorded as time series in units of $1.3 \times 10^{-6}$ $m^{-1}$ $sr^{-1}$.

Moreover, it is more convenient to use these numbers and it allows for shorter numbers which is very helpful for display purposes, so we would prefer to leave $10^{-6}$. But we have scaled the backscatter values by 1.3 in all plots and places in the text concerned. So now it is displayed in units of $10^{-6}$ $m^{-1}$ $sr^{-1}$ which is common practice in the literature. To further improve readability, we added to the caption of Figure 3:

(a) Scatter plot for Cabauw, with backscatter distribution at 12 km visibility, indicating a peak at $10^{-0.3}$ ×$10^{-6}$ $m^{-1}$ $sr^{-1}$.

This should make it quite easy for the reader to get the backscatter value should he/she wish so.

Figure 2c seems identical to a photo in Knoop et al. (2021). Please indicate source and license to reproduce it.

Response: The photo shows the lidar and met mast that measured the data for this paper. Knoop is co-author of this current paper and the data stems from the same campaingn, the photo corresponds to, so that should be fine. Moreover, Knoop et al 2021 is published under Attribution 4.0 International (CC BY 4.0), which means that things can be reproduced without permission, as long the initial source is mentioned. So a simple solution is = to put an explicit reference in the figure caption, which we have done now.

204-205 "A typical value for ð¼ has been empirically determined as 1.4 for visibilities between 6 and 20 km (Nebuloni et al., 2005), which is adopted here." Yet, on line 252-253 Angstrom exponent is changed to 2.0. Please give some more references to justify the selected Angstrom exponent and lidar ratio. At least Baars et al. (2016) and Illingworth et al. (2015) give some values for a few aerosol types, but there are probably better (and more recent) references.

Response: Yes we agree that is a bit confusing. I have moved the phrase from line 204 to line 252.This is because it is irrelevant there since at that point only equation (8) is presented. It makes more sense to present the value of the Angstroem coefficient where Eq. 8 is actually used, which is at line 252.

Line 253 reads now:

A typical value for the Ångström coefficient has been empirically determined as 1.4 for visibilities between 6 and 20 km (Nebuloni et al., 2005). This value largely overestimated visibilities from the visiometer. Increasing the Ångström coefficient to 2.0, associated with a finer, more continental aerosol dominating the backscatter, improved the fit considerably and the result is shown in Fig. 4a

209 Please check "Figure 3 3 shows"

Response:

Corrected to: Figure 3 shows

225-226 "The nonlinearity of the visibility with backscatter could be attributed to different contributions to the average aerosol size distribution (Curcio et al., 1958)." I don't quite understand what are the "different contributions" here, please clarify.

Response:

The point here was that the scatter plot already demonstrates that the ratio between visibility (i.e. inverse of the extinction coefficient) and the backscatter coefficient is not strictly constant. Since the backscatter strongly depends on the aerosol type, in particular size, this suggests that different aerosol types cause the different slopes.

At this point in the paper, however, I think talking about aeorosol properties already deems unnecessary and may confuse the reader, so we have rephrased the statement at l 225 to:

There appear to be two modes: A correlation with relatively flat slope for high visibilities, where data density is highest, and a steeper mode for visibilities below ~20 km. The curvature in Figure 3, similar to measurements at other locations (Fenn 1966) suggests a linear relationship of visibility and backscatter only over a limited range, as opposed to the relationship between visibility and extinction coefficient (Eq. 1). This implies that the lidar ratio (Eq. 6), is constant only for a limited range of visibility (or backscatter).

Moreover, at l 257 we changed

The scatter of the data (Figs. 3a and b) translates into mismatches between the timeseries of backscatter and visibility.

To

The scatter of the data and the nonlinear relationship between backscatter and visibility (Figs. 3a and b) translates into mismatches between the timeseries of backscatter and visibility.

**Figure 3: Please plot backscatter on logarithmic scale without the scaling factor.**

Response:

Please see related comment above for 180-181

**265-269 Please provide a literature overview of lidar-retrieved Angstrom exponent and lidar ratio at Cabauw and Pershore, or similar environments, if measurements are not available for these sites.**

Response:

There is indeed a measurement network, AERONET, that could be used to provide these parameters for certain sites and time intervals. We have made corresponding changes in the paragraph as follows:

Although the Ångström exponent does vary over short periods of time (hours to days), it does so in a confided manner. For certain sites, including Cabauw, Ångström exponents are available from the aerosol optical depth (AOD) product from the Aerosol Robotic Network (AERONET, https://aeronet.gsfc.nasa.gov/new_web/data_description_AOD_V2.html, last accessed 28/07/2022). For a few sites, lidar ratios are measured within the Portable Raman Lidar Network (PollyNet, Baars et al., 2016). For Cabauw, although none of these are available for autumn and winter 2018 from both networks, the Ångström exponent retrieved within AERONET at Cabauw

varies usually between 0.1 and 2.0 over the course of any given day and is bounded by these limits over the course of a year. This makes $S = 70$ sr, $\alpha = 2$ more likely than $S = 28$ sr, $\alpha = 2.6$. Figure S1 shows an example where time series of Ångström exponent at Cabauw from the AERONET (O'Neil et al., 2001) were used to improve the agreement with the visiometer especially for lower visibilities, while the agreement for the higher visibilities reduced, since a constant lidar ratio data was assumed.

While Method A is an interesting exercise, it is questionable whether it would be practical enough in obtaining a general transfer function between lidar backscatter and visibility. It could be feasible where monitoring networks such as AERONET exist that yield secondary parameters, including time series of the Ångström coefficient.

Supplementary material:

(a)

(b)

[Figure]

Figure S1. Linking lidar backscatter from 39 m agl to visibility for Cabauw using Method A. The visibility sensor data are from 40 m agl. The lidar ratio used is 70 sr. (a) A constant Ångström exponent of 2.0 was assumed. (b) Same as in (a) but coinciding Ångström exponent time series from the AERONET station at Cabauw were used, which were resampled to 10 min temporal resolution, as the lidar backscatter and the visiometer data.

Added reference:

O'Neill, N.T., O.Dubovik, O., and Eck, T. F.: A modified Angstrom coefficient for the characterization of sub-micron aerosols, Appl. Opt., 40, 2368-2375, 2001.

279 "The lidar backscatter coefficient can be quite dynamic, changing by several factors within minutes." Please specify which factors.

Response:

We have rephrased this sentence to:

The lidar backscatter coefficients at both was observed to fluctuate by up to 5 times within 10 minutes. To assess whether the 10 min averaging window caused any deterioration of the correlation, selected series of backscatter were offset by up to 5 minutes before averaging, with no significant effect on the correlation with the 10-min visibility time series.

To make clear that the data are measured at 1s-resolution, at l 183 we changed the phrase:

Both wind and visibility data are averaged over 10 min long periods.

To

Both backscatter and wind data are measured every second for a given height, but are averaged over 10 minute periods

Figure 5 and 6 captions: please define "BS".

Response:

I am sorry for that. That was supposed to mean backscatter. BS is changed to backscatter.

Moerover, for better readability and guidance to the reader we have added in caption of Figure 5:

(…) and the transfer function (black), to be read as $log(V^{-1}) = -3.724 + 1.291x$, where $V$ is visibility, $x = log(\beta)$, $log$ depicts the decadic logarithm.

309-310 "general observations of a vertically weakly exponential decrease in lidar signal strength (hence backscatter) that becomes significant above ~100 m agl." Is this due to lack of range correction in the backscatter retrieval?

Response: This is completely unrelated. Again, range correction is not applicable to our data, see discussion in first comment.

What is meant by this is that the backscatter, depending on the stability of the atmosphere, more or less decays with that trend, as an intrinsic property of the boundary layer. This can be seen from experience dealing with different wind lidar data sets from different locations ("general observation") and also from other lidars, such as Caliop. Here is an example from Caliop data:

[Figure]

To make this more clear to the readers we have expanded the sentence and made the following modifications:

(…) the visibilities computed from the fitted transfer functions of these correlations, show a possible slight upwards trend with increasing height (Fig. 8). It appears that visibility below ~80 m agl varies only little with height. This is expected, since, at least in unstable and neutral atmospheric conditions, no significant aerosol stratification at these low heights would be anticipated. It is also in line with other, long-term ZX300 lidar observations from various sites across the globe, suggesting a vertically weakly exponential decrease in lidar signal strength (hence backscatter) that becomes significant above ~200 m agl.. Although the vertical resolution is smaller than of most ground based lidar, these vertical trends can also be identified in data from the Caliop satellite borne aerosol lidar (Winkler et al., 2013). The temporal mean of the visibilities between 4 km and 19.5 km

We also changed ~100 m to ~ 200 m, as this is more in line with what we observe.

Reference added:

Winkler, D. M., Tackett, J. L., Getzewich, B. J., Liu, Z., Vaughan, M. A., and Rogers, R. R.: The global 3-D distribution of tropospheric aerosols as characterized by CALIOP, Atmos. Chem. Phys., 13, 3345–3361, https://doi.org/10.5194/acp-13-3345-2013, 2013.

345-346 "A backscatter minimum around July has been measured with different CW wind lidars in other locations in the Northern Hemisphere." Please add reference.

Response:

These are internal data and not published. To make this clear, the above phrase has been changed to:

A backscatter minimum around July has been measured with different ZX300 CW wind lidar systems at the ZX Lidars site near Ledbury, UK, but also other locations in the Northern Hemisphere (Scott Wyle, ZX Lidars 2022, personal communication).

**433-438 Are there any other studies that report similar seasonality for backscatter?**

Response:

Yes, there are. We have added few references. Digging into the source mechanisms is out of scope for this study, but interesting and could be tackled in another paper. We added:

Seasonality of atmospheric aerosol extinction, scattering and backscatter coefficients have been measured by others, such as with ground-based in situ measurements and lidar, including the arctic (Schmeisser et al., 2018; Shibata et al., 2018) and Spain (Sicard, et al., 2010; Navas-Guzmán et al., 2013). Mean extinction coefficients over several regions were measured with the Cloud-Aerosol Lidar with Orthogonal Polarization (CALIOP, Koffi et al., 2012). Consistent with the present finding, the mean extinction coefficient over Western Europe had a minimum in the summer. Schmeisser et al. (2018) found that in some locations around the Arctic Circle, maximum aerosol scattering occurred during spring and winter, whilst in other locations maximum extinction was measured during the summer months. This advocates as likely drivers of seasonality of aerosol backscatter regionally different mechanisms as well as the transport of aerosols into the region of measurement, such as from man-made sources (e.g., sulphates and soot particles, Shibata et al., 2018) or natural sources, including Sahara dust (Sicard et al., 2010; Navas-Guzmán et al., 2013) and sea salt (Koffi et al., 2012).

We have added the following references:

Navas-Guzmán, F., Bravo-Aranda, J. A., Guerrero-Rascado, J. L., Granados-Muñoz, M. J., and Alado s-Arboledas, L., Statistical analysis of aerosol optical properties retrieved by Raman lidar over Southeastern Spain, Tellus B: Chem. Phys. Meteorol., 65, doi: 10.3402/tellusb.v65i0.21234, 2013.

Koffi, B., Schulz, M., Breon, F-M., Griesfeller, J., Winker, D. M., et al., Application of the CALIOP layer product to evaluate the vertical distribution of aerosols estimated by global models: AeroCom phase I results. J. Geophys. Res.: Atmospheres, 117, D10201, doi:10.1029/2011JD016858, 2012.

Schmeisser, L., Backman, J., Ogren, J. A., Andrews, E., Asmi, E., Starkweather, S., Uttal, T., Fiebig, M., Sharma, S., Eleftheriadis, K., Vratolis, S., Bergin, M., Tunved, P., and Jefferson, A.: Seasonality of aerosol optical properties in the Arctic, Atmos. Chem. Phys., 18, 11599–11622, https://doi.org/10.5194/acp-18-11599-2018, 2018.

Shibata, T., Shiraishi, K., Shiobara, M., Iwasaki, S., & Takano, T. (2018). Seasonal variations in high Arctic free tropospheric aerosols over Ny-Ålesund, Svalbard, observed by ground-based lidar. Journal of Geophysical Research: Atmospheres, 123, 12,353–12,367. https://doi.org/10.1029/2018JD028973

455-457 and 462-464 See e.g. Illingworth et al. (2015) and Baars et al. (2016) for range of values associated with different aerosol types. Please also check if you can find better references on the topic.

Response:

The references in our paper you are mentioning are concerned with Mie theory to explain the discrepancy between backscatter and extinction coefficient on a fundamental level. These references, at least those from the 1960ies, present quite fundamental, pioneering results so we would prefer to keep those where they are. Illingworth et al. (2015) and Baars et al. are very interesting papers, would be very helpful earlier in the paper where we discuss method A to retrieve visibility from backscatter only without the need for secondary collocated instrumentation. To this end, networks such as PollyNet could be a very useful addition to Aeronet and help to use wind lidars to estimate visibility by providing Angstroem exponents or depolarization ratios. We have added Baars et al., please see comment "265-269 Please provide a literature overview of lidar-retrieved Angstrom exponent" above

Added: ==using a visiometer== at l 482

Since the difference in transfer function is in all likelihood related to differences in the predominant, local aerosol SD and/or particle number density, this indicates that after calibrating the backscatter measured by the CW wind lidar (method B) ==using a visiometer==, the lidar could be used to measure visibility. The same applies to method A.

Added reference:

Baars, H., Kanitz, T., Engelmann, R., Althausen, D., Heese, B., Komppula, M., Preißler, J., Tesche, M., Ansmann, A., Wandinger, U., Lim, J.-H., Ahn, J. Y., Stachlewska, I. S., Amiridis, V., Marinou, E., Seifert, P., Hofer, J., Skupin, A., Schneider, F., Bohlmann, S., Foth, A., Bley, S., Pfüller, A., Giannakaki, E., Lihavainen, H., Viisanen, Y., Hooda, R. K., Pereira, S. N., Bortoli, D., Wagner, F., Mattis, I., Janicka, L., Markowicz, K. M., Achtert, P., Artaxo, P., Pauliquevis, T., Souza, R. A. F., Sharma, V. P., van Zyl, P. G., Beukes, J. P., Sun, J., Rohwer, E. G., Deng, R., Mamouri, R.-E., and Zamorano, F.: An overview of the first decade of Polly[NET]: an emerging network of automated Raman-polarization lidars for continuous aerosol profiling, Atmos. Chem. Phys., 16, 5111–5137, https://doi.org/10.5194/acp-16-5111-2016, 2016.

525-527 "For Cabauw, lidar backscatter derived visibility was found to be height dependent (Fig. 8), in line with the observation that under cloud free conditions backscatter from CW-wind lidar usually tends to slightly decrease with height in the lower part of the planetary boundary layer." Is this due to lack of range correction in the backscatter retrieval?

Response:

No, this just refers to the intrinsic vertical backscatter profile. Please see comment above on weakly exponentially decrease of backscatter.

Other changes made:

"Ångström coefficient" changed to "Ångström exponent"

Line 229: Added: "For Pershore, data for the same range of visibilities as for Cabauw are selected. "

Line 239: logarithmised changed to logarithmic

Changed "Sensor" to "Visiometer" in Fig. 4

Caption of Figure 4: Changed "Linking backscatter" to "Linking lidar backscatter"

L 311: Added:

For comparison, this was done using visibility values from the lidar backscatter and visibility values from the visiometer.

L317:

Changed:

The intercept for the Cabauw data is smaller than that for Pershore, whilst the slope is slightly steeper, which indicates that the annual average backscatter is below that of Cabauw for all visibilities considered here.

To

The difference to Cabauw in intercept and slope of the transfer function (Fig. 6a) indicates that the annual average backscatter is below that of Cabauw for all visibilities considered here.

L420:

It was found that limiting the data acquisition period practically decreases the variance of the backscatter distribution at a given visibility, i.e., has a similar effect as increasing the threshold (Eq. 9), at the expense of an increased risk of a poorer linear fit, which could explain an increase in MAE

L424:

Merged "Increasing the threshold above a certain level may have the same effect." into previous phrase as follows:

Limiting the data period further (or increasing the threshold above a certain level) may decreases the number of data and hence the goodness of the fit, at which point the MAE may increase.

L425: Deleted as redundant:

"In practice, therefore, increasing the threshold of the fit has the same effect as limiting the data period for fitting (improves MAE), at least for the limited data set available here."

L425: Changed:

For a very large data set (e.g. 10 years of data), however, matching the data period for the fit to that for the prediction could possibly be more beneficial than a simple thresholding of the 2D-histogram.

To

For a very large data set (e.g. 10 years of data), however, matching the data period for the fit to that for the prediction could possibly be beneficial to predict more accurate visibilities.

L525:

Changed:

The result by Tworney and Howell (1965) suggests that the use of monochromatic light contributes to the spread observed in the correlation between backscatter and visibility with a factor of ~2, but it does not explain a systematic offset.

To

The result by Tworney and Howell (1965) suggests that, due to this effect, the spread of backscatter for a given visibility (Figs. 5, 6, 7, and 9) is up to twice as high as it would be for a measurement with polychromatic light.

The systematic offset between two sites too is likely caused by a different local aerosol SD to which the backscatter is more sensitive than forward scatter at the angular ranges used in the visibility sensors.

L554 Changed:

The strong concentration of aerosols from both road and sea traffic and industrial air pollution downwind of Rotterdam and Rotterdam harbour may explain the larger average backscatter at Cabauw and hence the difference in intercept of the transfer functions between the two sites. The difference in slope is likely dominated by the difference in the lidar ratio, i.e., due to different dominating aerosol type(s) at the two sites.

To

The strong concentration of aerosols from both road and sea traffic and industrial air pollution downwind of Rotterdam and Rotterdam harbour may explain the larger average backscatter at Cabauw (Fig. 10) and hence the difference in the transfer functions between the two sites (Figs. 5a and 6b). The difference in slope of the transfer functions is likely dominated by the difference in the lidar ratio, i.e., due to different dominating aerosol type(s) at the two sites.

L559:

Condensed phrase from:

However, other diurnal mechanisms most certainly will affect aerosol type and number density, and hence lidar backscatter, such as meteorological processes, for example boundary layer mixing processes (Stanier et al., 2004).

To

However, other diurnal mechanisms, such as boundary layer mixing processes (Stanier et al., 2004), most certainly will affect aerosol type and number density, and hence lidar backscatter.

Added reference, l 585:

Fenn, Robert W. (1966). *Correlation Between Atmospheric Backscattering and Meteorological Visual Range. ao/5/2/ao-5-2-293.pdf, 5(2), 293–0.* doi:10.1364/AO.5.000293

L589 Changed:

Directly relating backscatter to visibilities was found less practical due to the need for additional input parameters.

To

Directly relating backscatter to visibilities was found practical only in cases where additional input data to the backscatter, that is, lidar ratio and Ångström exponent, are available.

L594 Changed:

This can be explained by different aerosol types and size distributions at play for different backscatter coefficients

To

This can be explained by different aerosol types and size distributions at play for different backscatter coefficients at a given location over the course of time (Fenn et al., 1966).

L596 Changed:

Differences in local dominant aerosol type between two locations lead to differences in extinction-to-backscatter ratio between the two sites and thus differences in the transfer function.

To

Related to this, differences in local dominant aerosol type between two locations (even at a given time) lead to differences in extinction-to-backscatter ratio between the two sites and thus differences in the transfer function.

L603 Added: of the lidar

References

Baars, H., Kanitz, T., Engelmann, R., Althausen, D., Heese, B., Komppula, M., Preißler, J., Tesche, M., Ansmann, A., Wandinger, U., Lim, J.-H., Ahn, J. Y., Stachlewska, I. S., Amiridis, V., Marinou, E., Seifert, P., Hofer, J., Skupin, A., Schneider, F., Bohlmann, S., Foth, A., Bley, S., Pfüller, A., Giannakaki, E., Lihavainen, H., Viisanen, Y., Hooda, R. K., Pereira, S. N., Bortoli, D., Wagner, F., Mattis, I., Janicka, L., Markowicz, K. M., Achtert, P., Artaxo, P., Pauliquevis, T., Souza, R. A. F., Sharma, V. P., van Zyl, P. G., Beukes, J. P., Sun, J., Rohwer, E. G., Deng, R., Mamouri, R.-E., and Zamorano, F.: An overview of the first decade of PollyNET: an emerging network of automated Raman-polarization lidars for continuous aerosol profiling, Atmos. Chem. Phys., 16, 5111–5137, https://doi.org/10.5194/acp-16-5111-2016, 2016.

Hopkin, E., Illingworth, A. J., Charlton-Perez, C., Westbrook, C. D., and Ballard, S.: A robust automated technique for operational calibration of ceilometers using the integrated backscatter from totally attenuating liquid clouds, Atmos. Meas. Tech., 12, 4131–4147, https://doi.org/10.5194/amt-12-4131-2019, 2019.

Illingworth, A. J., Barker, H. W., Beljaars, A., Ceccaldi, M., Chepfer, H., Clerbaux, N., Cole, J., Delanoë, J., Domenech, C., Donovan, D. P., Fukuda, S., Hirakata, M., Hogan, R. J., Huenerbein, A., Kollias, P., Kubota, T., Nakajima, T., Nakajima, T. Y., Nishizawa, T., Ohno, Y., Okamoto, H., Oki, R., Sato, K., Satoh, M., Shephard, M. W., Velázquez-Blázquez, A., Wandinger, U., Wehr, T., and van Zadelhoff, G.-J.: The EarthCARE Satellite: The Next Step Forward in Global Measurements of Clouds, Aerosols, Precipitation, and Radiation, Bull. Amer. Meteor. Soc., 96, 1311–1332, https://doi.org/10.1175/BAMS-D-12-00227.1, 2015.

Knoop, S., Bosveld, F. C., de Haij, M. J., and Apituley, A.: A 2-year intercomparison of continuous-wave focusing wind lidar and tall mast wind measurements at Cabauw, Atmos. Meas. Tech., 14, 2219–2235, https://doi.org/10.5194/amt-14-2219-2021, 2021.

Kotthaus, S., O'Connor, E., Münkel, C., Charlton-Perez, C., Haeffelin, M., Gabey, A. M., and Grimmond, C. S. B.: Recommendations for processing atmospheric attenuated backscatter profiles from Vaisala CL31 ceilometers, Atmos. Meas. Tech., 9, 3769–3791, https://doi.org/10.5194/amt-9-3769-2016, 2016.

Pentikäinen, P., O'Connor, E. J., Manninen, A. J., and Ortiz-Amezcua, P.: Methodology for deriving the telescope focus fu

nction and its uncertainty for a heterodyne pulsed Doppler lidar, Atmos. Meas. Tech., 13, 2849–2863, https://doi.org/10.5194/amt-13-2849-2020, 2020.

Vakkari, V., Manninen, A. J., O'Connor, E. J., Schween, J. H., van Zyl, P. G. and Marinou, E.: A novel post-processing algorithm for Halo Doppler lidars, Atmos. Meas. Tech., 12(2), 839–852, doi:10.5194/amt-12-839-2019, 2019.

---

## Author Comment (AC2)

**Response to comments of reviewer #2**

The authors would like to thank the reviewer for the time and effort taken to review this manuscript, which is very much appreciated.

Following the response to the other reviewer (reviewer '#1), the authors would like to point out that we believe there was a slight misunderstanding with regards to the validity of the method used (retrieval of backscatter), which negatively biased the review criteria mark of reviewer 1 and possibly reviewer 2.

We have made substantial improvements to the manuscript that will hopefully help to show that the method to retrieve backscatter is different to pulsed lidars and ceilometer, but no less valid. For more details, please see response to comments of reviewer #1.

We furthermore made improvements to readability of the manuscript by changing the scaling of backscatter value in the graphs as requested by the reviewer #1 and by improving the structure of the discussion and result sections.

Changes made to the manuscript are highlighted in yellow.

The manuscript suggests correlation between visibility measurements from in-situ visiometers and backscatter coefficient measurements from a continuous-wave wind Lidar. Datasets from two measurement campaigns are used, one in **Cabauw** (Netherlands) and one in **Pershore** (UK). The study falls into the scope of AMT. Yet, there are important differences between visibility from CW wind lidar and visiometers, arising from the different aerosol properties. Also, calibration of CW wind lidar against a visibility sensor in a similar mean aerosol scene area to the one of its intended use is necessary, creating limitations to future applications.

At L 104-106 when describing the CW wind lidars, it is said that "the return signal is not sensitive to atmospheric extinction, but is practically governed by the backscatter coefficient only. This leaves the backscatter coefficient as the most obvious proxy of visibility of a CW lidar." Thought at L43 it has been mentioned that "by measuring light extinction σ, MOR can be derived". Basically, it looks like the most important parameter is overlooked. Could you give more details on that?

Response:

Yes, this is the "standard" way of doing it. This paper presents an alternative way because cw wind lidar is not sensitive to extinction but only backscatter.

For clarity we added the following phrase on l 82:

The question arises, whether MOR can be retrieved from only one of the two parameters, backscatter coefficient or extinction coefficient.

Added in l 50:

This is also the height at which state of the art visibility sensors, or visiometers, determine MOR, using Eq. 1.

We have added a small calculation that illustrates that a cw wind lidar signal is not sensitive to extinction:

The paragraph now reads:

As opposed to pulsed aerosol lidar described above, CW wind lidar has a lower measurement range. In addition, CW wind lidar operates in the short wave infrared region close to 1550 nm, which is a factor of 1.5 to 3 longer than for typical aerosol lidar and ceilometer systems (Werner et al., 2005; Gasteiger et al., 2011; Navas-Guzmán et al., 2013; Shibata et al., 2018). Compared with pulsed aerosol lidar, at normal working ranges (up to 300 m), the return signal of a CW wind lidar is thus not sensitive to atmospheric extinction, but is practically governed by the backscatter coefficient only. This is illustrated in the following example. Assuming a common visibility of 10 km (slightly hazy), the maximum extinction (i.e. at range 300 m) would only be $e^{-600\,m\,5\times10^{-5}m^{-1}} = 0.97$. This leaves only the backscatter coefficient (henceforth termed backscatter) as the most obvious proxy of visibility of a CW wind lidar.

L538-539 "This can be explained by different aerosol types and size distributions at play for different backscatter coefficients" I find it hard to understand the grammar of this sentence.

Response:

The phrase has been amended as follows:

This non-linearity suggests the lidar ratio is constant over a limited range of backscatter values only, which in turn is consistent with the contribution of different aerosol types and size distributions for different backscatter coefficients at a given location over the course of time (Fenn et al., 1966).

I think that the small agreement between CW Doppler lidar and visibility sensor measurements, mainly for Pershore, should be mentioned in the conclusions and briefly explain the reason of these differences.

Response:

L 509: Deleted following phrase since a reasonable explanation is provided further down from l 509:

While we do not have an explanation for it, it also appears to be related to the site-specific average aerosol type.

Amended and added the following phrase at l 525:

This also implies that the use of polychromatic light yields a backscatter intensity less dependent on the aerosol SD than the highly monochromatic light of a coherent wind lidar.

The result by Tworney and Howell (1965) suggests that the use of monochromatic light contributes to the spread observed in the correlation between backscatter and visibility with a factor of ~2, but it does not explain a systematic offset. The latter is more likely to be caused by a different local aerosol SD to which the backscatter is more sensitive than forward scatter at the angular ranges used in the visibility sensors.

To

This also implies that the use of polychromatic light yields a backscatter intensity less dependent on the aerosol SD than the highly monochromatic light of a coherent wind lidar. The result by Tworney and Howell (1965) suggests that, due to this effect, the spread of backscatter for a given visibility (Figs. 5, 6, 7, and 9) is up to twice as high as it would be for a measurement with polychromatic light. The magnitude of the spread is likely also a function of the variety of aerosol types and hence SD present over the data acquisition period, which increases non-uniqueness of the relation between visibility and backscatter (Fenn 1966). This suggests that, relative to Cabauw, Pershore experienced a higher variety in aerosol type and SD.

In the Conclusions we added (l 598):

For larger ranges of visibility and backscatter coefficients, the correlation was found to be less linear. The method deems, therefore, practical only over a limited parameter range. This implies that the lidar ratio is constant over a limited range of backscatter values only. In addition, the results indicate a spread of backscatter values for a given visibility, with the spread being dependent on the location. The spread corresponds to a nonunique relation between visibility and backscatter. Both the nonlinearity and the nonuniquenes are linked to the contribution of a variety of aerosol types and size distributions for a given backscatter coefficient at a given location over the data acquisition period (Fenn 1966). Related to this, differences in local dominant aerosol type between two locations (even at a given time) lead to differences in lidar ratio between the two sites and thus differences in the transfer function.

Concerning the site specific differences, it should be mentioned that if backscatter from other types of instruments (e.g.ceilometers) was used, the same differences between the two sites would have arisen and also provide an example of correspondingly data. The site specific differences are very important and every site will present different aerosol scene and properties.

Response:

That is a good point that I already have addressed in response to reviewer #1. I haven't put it in the manuscript yet, but I will do.

We added at l609:

Both nonlinearity and nonuniqueness are independent of the setup used to measure backscatter (e.g., CW lidar, pulsed lidar, flash light, Curcio and Knestrick, 1958; Doherty et al., 1999; Werner et al., 2005, p. 172).

Since the study assesses if backscatter from CW wind lidar can be used to retrieve visibility, a conclusion about seasonality observed for backscatter in the two sites (Fig 10), along with the MOR connection to this seasonality, would be really helpful for the reader.

Response:

Not directly related to your request, but still relevant: Following the request of reviewer #1, we have already added in the discussion other paper that found seasonality in aerosol scattering and backscattering using other techniques.

On l 482 we added:

However, depending on the year, the result suggests uncertainties up to a few kilometres if a transfer function for a given month was used to predict visibility for the same month of a different year.

In the conclusions we added:

For the 2-year data set used here, selecting a subset of the data (season, month etc.) did not improve the accuracy of the transfer function, i.e., accuracy of predicting visibility for the corresponding subset of the same year or a different year. Data sets acquired over more than two years may improve the accuracy of the transfer function

Have you checked what happens if you use Ct=2% in eq. (1)? Eg. (8) would change to $4/(\beta(\pi)S(\lambda_1/\lambda_0)^\alpha)$. Would this have an important effect on the results?

Response:

It wouldn't affect the conclusions, to ideally have secondary measurements of S and $^\alpha$. It would increase the lidar derived visibility by 30% , which does not automatically mean it would be 30% higher than the visiometer readings, since the misfit varies along the x-axis, plus S and $^\alpha$ where not found by fitting the visometer reading, which means the fit might actually improve at places, but overall the visometer readings would be overestimated by probably at least 10%. The overestimated lidar visibilities could be "corrected" by increasing S and/or $^\alpha$, so:

by a) increasing the lidar ratio by up to 10% or more, or

b) the Angstrom exponent by up to ~13%, or

c) a combination of both, obviously with lower factor of increase

Options a) and b) would lead to parameters a bit high, but not impossible (Doherty et al., 1999, see Bibliography in manuscript), c) would entail lidar ratio and Angstrom coefficient to be fully within average values at Cabauw.

We have added more information on lidar ratio and Angstrom exponents at Cabauw and how to get them in the manuscript:

Although the Ångström exponent does vary over short periods of time (hours to days), it does so in a confided manner. For certain sites, including Cabauw, Ångström exponents are available from the aerosol optical depth (AOD) product from the Aerosol Robotic Network (AERONET, https://aeronet.gsfc.nasa.gov/new_web/data_description_AOD_V2.html, last accessed 28/07/2022). For a few sites, lidar ratios are measured within the Portable Raman Lidar Network (PollyNet, Baars et al., 2016). For Cabauw, although none of these are available for autumn and winter 2018 from both networks, the Ångström exponent retrieved within AERONET at Cabauw varies usually between 0.1 and 2.0 over the course of any given day and is bounded by these limits over the course of a year. This makes $S = 70$ sr, $\alpha = 2$ more likely than $S = 28$ sr, $\alpha = 2.6$. Figure S1 shows an example where time series of Ångström exponent at Cabauw from the AERONET (O'Neil et al., 2001) were used to improve the agreement with the visiometer especially for lower visibilities, while the agreement for the higher visibilities reduced, since a constant lidar ratio data was assumed.

Also, lidar ratio S is considered constant, but as also described in the manuscript, presents strong variability depending on the aerosol type. This should also be mentioned in the conclusions (L 537-544) along with the "less linear correlation".

Response:

This has been added now in the conclusions as follows:

In the Conclusions we added (I 598):

For larger ranges of visibility and backscatter coefficients, the correlation was found to be less linear. The method deems, therefore, practical only over a limited parameter range. This implies that the lidar ratio is constant over a limited range of backscatter values only.

Other changes made by the authors:

Removed the following phrase from l543, as it is redundant:

More long-term tests are needed to assess to which extend the calibration needs to be repeated or if it needs to be repeated if the location is kept fixed.

L546:

Added:

==Furthermore, a landfill, located ~800 m to the southwest of the lidar location, could also produce enhancements in Diesel aerosol from the operated machinery and aerosols common to landfills (Nair 2021).==

Added reference:

Nair AT. Bioaerosols in the landfill environment: an overview of microbial diversity and potential health hazards. Aerobiologia (Bologna). 2021;37(2):185-203. doi: 10.1007/s10453-021-09693-9. Epub 2021 Feb 4. PMID: 33558785; PMCID: PMC7860158.

L615:

Changed the phrase:

Going forward, it might be useful to confirm or refine this conclusion by measuring at more sites globally and categorize them into sites with similar predominant mean aerosol SD. Obtaining visibility data from more sites is desirable to test how site specific the transfer function is and how comparable it is between similar environmental settings.

To

Going forward, it might be useful ==to acquire transfer functions== at more sites globally and categorize them into sites with similar predominant mean aerosol ==size distributions==. Obtaining visibility data from more sites is desirable to test how site specific the transfer function is and how comparable it is between similar environmental settings.